# Organic soil CO₂ balance in drained and undrained hemiboreal forests

Aldis Butlers[1], Raija Laiho[2], Andis Lazdiņš[1], Thomas Schindler[3], Kaido Soosaar[3], Jyrki Jauhiainen[2], Arta Bārdule[1], Muhammad Kamil-Sardar[3], Ieva Līcīte[1], Valters Samariks[1], Andreas Haberl[4], Hanna Vahter[3], Dovilė Čiuldienė[5], Jani Anttila[2], Kęstutis Armolaitis[5]

[1]Latvian State Forest Research Institute (Silava), Salaspils, 2169, Latvia
[2]Natural Resources Institute Finland (Luke), P.O. Box 2, Helsinki 00791, Finland
[3]Department of Geography, University of Tartu, Tartu, 51014, Estonia
[4]Michael Succow Foundation (partner in the Greifswald Mire Centre), 17489 Greifswald, Germany
[5]Department of Silviculture and Ecology, Lithuanian Research Centre for Agriculture and Forestry, Kėdainiai distr., 58344, Lithuania

*Correspondence to*: Aldis Butlers (aldis.butlers@silava.lv)

**Abstract**. Drainage of organic soils is associated with increasing soil carbon dioxide ($CO_2$) efflux, which is typically linked to losses in soil C stock. In previous studies, drained organic forest soils have been reported as both $CO_2$ sinks and sources depending on, e.g., soil nutrient and moisture regime. However, most of the earlier research was done in the boreal zone, and both the magnitude of $CO_2$ efflux and the impact of soil moisture regime on soil C stock are likely to vary across different climatic conditions and ecosystems, depending further on vegetation. A two-year study was conducted in hemiboreal forest stands with nutrient-rich organic soil (including current and former peatlands) and a range of dominant tree species (black alder, birch, Norway spruce, Scots pine) in the Baltic states (Estonia, Latvia, Lithuania). In this study, we analyzed the $CO_2$ balance of organic soil in drained (19) and undrained (7) sites. To assess the $CO_2$ balance, soil respiration was measured along with evaluation of $CO_2$ influx into the soil through aboveground and belowground litter. To characterize the sites and factors influencing the $CO_2$ fluxes, we analysed soil temperature, soil water-table level, and physical and chemical parameters of soil and soil water. On average, close-to-neutral soil $CO_2$ balance ($+0.45 \pm 0.50$ t $CO_2$-C ha⁻¹ year⁻¹) was observed in drained sites dominated by black alder, birch, or Norway spruce, while drained Scots pine sites showed soil $CO_2$ removals with a mean rate of $+2.77 \pm 0.36$ t $CO_2$-C ha⁻¹ year⁻¹. In undrained birch- and spruce-dominated sites, soil functioned as mean $CO_2$ sink at $+1.33 \pm 0.72$ t $CO_2$-C ha⁻¹ year⁻¹, while the undrained black alder stands showed an uncertain $CO_2$ balance of $+1.12 \pm 2.47$ t $CO_2$-C ha⁻¹ year⁻¹. Variation in the soil $CO_2$ balance was related to soil macronutrient concentrations and pH: forest types characterized by lower nutrient availability showed greater soil $CO_2$ sink.

## 1 Introduction

Soil in peatlands, characterised by its high content of partially decomposed plant matter, is a major terrestrial organic carbon (C) stock, estimated to range from 504 to 3000 Gt C (Scharlemann et al., 2014). Although northern peatlands make up only 2-4% of the global land area, they contain a substantial amount of soil C, estimates ranging from around 500 to 1,055 Gt C (Nichols and Peteet, 2019; Yu, 2012), highlighting the significance of these lands in the global C budget. About 28% of the pristine (undrained) peatlands globally are inherently covered by forest (Zoltai and Martikainen, 1996), and those peatland forests in the boreal biome can accumulate C into the soil at similar rates to non-forested peatlands; the faster decomposition rates observed in peatland forests (Beaulne et al., 2021) can be compensated by higher litter inputs (Straková et al., 2010). To enhance tree growth, peatland drainage for forestry has been common in the past. Drainage facilitates oxygen access to deeper peat layers, thereby promoting tree root survival and function, but also the mineralization of organic matter and the release of C into the atmosphere in the form of $CO_2$. Therefore, the conservation of organic soil C stocks in managed current and former peatlands has attracted attention in the context of climate change.

The approximately 13 million ha of forestry-drained organic soils in Europe have been estimated to emit 17 million tons of $CO_2$ per year (European Environment Agency, 2023). Despite the temperate zone being characterized by wide climatic gradients, currently, only a single default emission factor (EF) by the Intergovernmental Panel on Climate Change (IPCC) is available for the entire temperate zone, to which the Baltic states correspond according to the IPCC (Hiraishi et al., 2013). The EF was developed using study results from 8 drained sites (Hiraishi et al., 2013), which were published in 5 articles (Von Arnold et al., 2005; Glenn et al., 1993; Minkkinen et al., 2007; Yamulki et al., 2013). These studies employed different $CO_2$ estimation methods, complicating the comparability of the aggregated results (Jauhiainen, 2019; Jauhiainen et al., 2023). None of the sites are in the Baltic states. Given that the soil emissions in the boreal zone are smaller than in the temperate zone (Jauhiainen et al., 2023), and Baltic states are situated in the hemiboreal vegetation zone (Ahti et al., 1968) – in transition between the temperate and boreal zones – the IPCC's default temperate EF may not be suitable for application in this region. The same issue arises on a broader geographic scale, where the use of unharmonized country-specific and default EFs creates challenges in comparing the estimated emissions both within and across different countries and climate zones. Discrepancies in $CO_2$ emissions between regions can be expected due not only to climate gradient (Ojanen et al., 2010) but also to site productivity (Janssens et al., 2001). While higher ecosystem productivity is associated with increased soil respiration rates (Janssens et al., 2001) it also facilitates higher $CO_2$ influx through litter (Krasnova et al., 2019). Therefore, EFs are most appropriately used when applied to areas with similar environmental conditions, rather than being limited by national boundaries or applied too broadly across diverse geographic regions.

In the Baltic states (Estonia, Latvia, and Lithuania), organic soils (Eggleston et al., 2006) are current or former peatlands where a peat layer is still identifiable, or, due to high decomposition, no longer meets the typical characteristics of peat. However, these soils by definition contain at least a 20 cm thick layer rich in organic matter (organic layer). In the region, the total area of drained organic forest soils is reported to be 0.8 million ha, with estimated emissions of 1.8 million tons of $CO_2$ per year (Konstantinavičiūtė et al., 2023; Ministry of the Environment of Republic of Estonia, 2021; Skrebele et al., 2023). Thus, countries with a relatively small total land area yet a substantial proportion of organic soil can have a considerable role in organic soil management. This underscores the importance of acquiring precise estimates of the $CO_2$ emissions from organic forest soils in this region. However, despite the Baltic States being located next to each other and thus expectedly showing similar soil $CO_2$ emissions from comparable sites and land uses, the emission estimation approach is currently not harmonized as the countries use different EFs to estimate and report emissions (Konstantinavičiūtė et al., 2023; Ministry of the Environment of Republic of Estonia, 2021; Skrebele et al., 2023). According to National Greenhouse Gas Inventories submissions of 2023, the $CO_2$ emissions of drained organic forest soil in Estonia and Lithuania were estimated using the default EF provided by IPCC for the temperate zone (Calvo Buendia et al., 2019), while Latvia applied a country-specific EF. Due to similarities in biogeography, climate, and land-use practices, common EFs based on regionally representative, congregated data could be a better option.

A recent synthesis evaluated whether default IPCC EFs can be improved by compiling results from the most recent studies (Jauhiainen et al., 2023). Still, only modest and insignificant changes judging by confidence intervals (CI) of IPCC EFs could be introduced for the temperate climate zone as a whole, due to limited data (Jauhiainen et al., 2023). Although the general driving factors of $CO_2$ emissions are known, the number and geographical representation of studies on drained soils, particularly in the temperate zone, remain too limited for stratification of EFs based on local conditions. It is widely known that soil temperature is the primary factor influencing gross $CO_2$ emissions. However, while temperature can explain variations in emissions, it does not fully account for their magnitude. The extent of $CO_2$ emissions is affected by soil properties (Basiliko et al., 2007), incorporating complex interactions between a soil's physical, chemical, and biological characteristics, including the impact of the litter quality (Berger et al., 2010). The recent synthesis confirmed that key factors influencing the magnitude of $CO_2$ emissions include soil C concentration, carbon-to-nitrogen (C:N) ratio, and bulk density, as well as stand type

(Jauhiainen et al., 2023). Yet, the list likely remains quite incomplete, as individual studies in different site types using various methods can provide unharmonized results making it difficult to identify the relationships influencing $CO_2$ emissions. Varying and often insufficient reporting of study site conditions in previous studies (Jauhiainen, 2019) further hampers the ability to compile the results for effective synthesis and meta-analysis (Jauhiainen et al., 2023). This limitation may have hindered the identification of emission-impacting factors and the ability to quantify their relationships, underscoring the need for more localized studies to address these gaps, particularly in the hemiboreal vegetation zone which overlaps with the Cool Temperate Moist climate zone (Calvo Buendia et al., 2019) - a subregion of temperate zone as defined by the IPCC.

In the few studies on drained and undrained soil C or $CO_2$ balance conducted in the Baltic states, using both chamber and soil inventory methods, findings have been inconsistent (Butlers et al., 2022; Lazdiņš et al., 2024; Bārdule et al., 2022). Drained organic soils have been identified as both C sinks and sources, with no decisive conclusions reached regarding the factors driving such variation. Soil C loss of nutrient-rich organic soil has been identified using the soil inventory method (Lazdiņš et al., 2024) but not confirmed by the chamber method (Butlers et al., 2022; Bārdule et al., 2022) that, unlike the inventory method, targets the current situation (Jauhiainen et al. 2019). Given that nutrient-rich organic forest soils can make up to 72 % of total organic forest soils (Līcīte et al., 2019) and are associated with a higher risk of soil C loss, there is a need to enhance our understanding of the underlying drivers to improve the accuracy of $CO_2$ balance estimates in Baltic states. Therefore, studies using harmonised data collection methods are necessary, covering a variety of nutrient-rich forest soils, recording or monitoring essential environmental conditions, and accounting for both soil $CO_2$ efflux and influx from various litter sources.

This study aimed to quantify the soil $CO_2$ balance in hemiboreal forests with nutrient-rich organic soil and different dominant tree species. The research was carried out in 26 forest stands with organic soil in Estonia (EE), Latvia (LV), and Lithuania (LT), including both undrained and drained sites, over two years. We analysed soil $CO_2$ emissions as well as $CO_2$ influx by tree fine roots, ground vegetation (below- and aboveground) and fine foliar litter. We examined factors contributing to soil $CO_2$ balance by directly or indirectly influencing soil processes, such as site type, dominant tree species, soil and soil water properties, and soil water-table level (WTL). Mean annual soil $CO_2$ balance values were estimated for potential use as EFs. For this purpose, we also assessed whether the $CO_2$ fluxes differed between countries. To facilitate the use of the results in future syntheses or meta-analyses, the data used for $CO_2$ balance estimation have been openly published.

## 2 Materials and methods

### 2.1 Study sites

In total, 26 study sites (Figure 1) were established in stands dominated by black alder (*Alnus glutinosa* (L.) Gärtner), birch (*Betula pendula* Roth, *Betula pubescens* Ehrh.), Scots pine (*Pinus sylvestris* L.), or Norway spruce (*Picea abies* (L.) Karst.) of different ages (Table S1). The study sites included both drained (n=19) and undrained (n=7) organic soils (Eggleston et al., 2006), with the organic layer thickness ranging from 27 cm to over 2 meters, measured by a rod insertion. According to forest type classification, all sites were characterised as nutrient-rich based on ground vegetation and stand productivity (Bušs, 1981). Drained sites were represented by two site types: *Oxalidosa turf. mel.* (Oxalis), which has relatively higher nutrient availability (pH, macronutrients), and *Myrtillosa turf. mel.* (Myrtillus). Undrained sites were represented by the *Dryopterioso–caricosa* site type. Soil drainage status was determined based on the presence of drainage ditches within or along the respective forest compartment (a rectangular forest area of $50 \pm 25$ ha used as a management unit).

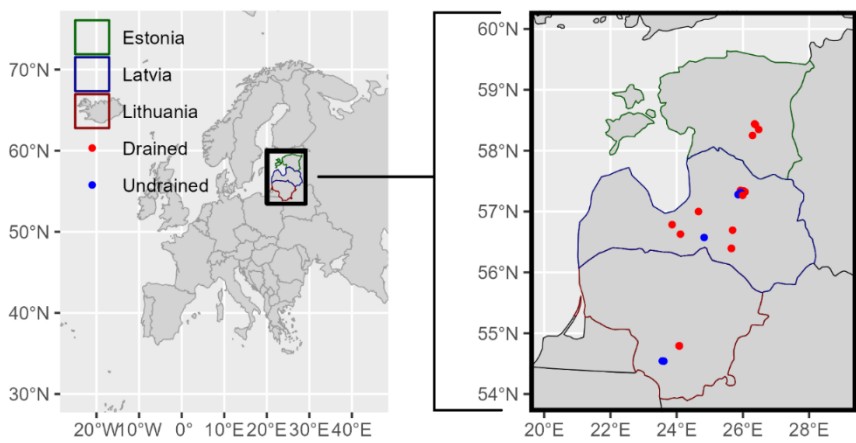

Figure 1: Locations of the study sites. Points indicate the locations of study site clusters.

Despite the greater variation in stand characteristics among the more numerous drained sites compared to the undrained sites, the two groups were overall comparable. (Table 1). The mean stand age of both groups was 74 years, with a range of 26-162 years for drained and 44-96 years for undrained sites. Average basal areas in turn were 27 for drained and 30 m$^2$ ha$^{-1}$ for undrained sites, respectively. More information on stand characteristics, including mean WTL and coordinates, is provided in Table S1. The projective cover of the most common ground vegetation species in the study sites is presented in Table S2. All comparisons will be done at group level, since pairwise comparison, even of closely located sites that belong to the same site type, is hampered by the inherent variation in soil characteristics (see Laiho and Pearson 2016).

Both groups represent historically naturally forested lands that have undergone active forest management. While the exact time since drainage is unknown, active drainage of the forests was initiated in the mid-19th century and resumed in the mid-20th century. Drainage ditches were dug to depths of 0.8 to 1.2 m, with distances between them ranging from 100 to 400 m, and they are maintained to remain functional (Zālītis, 2012).

Table 1: Range (minimum; maximum) of tree stand characteristics in the study sites.

| Parameter | Dominant tree species | | | | | | | |
| | Black alder | | Birch | | Pine | | Spruce | |
| | Drained | Undrained | Drained | Undrained | Drained | Undrained | Drained | Undrained |
| --- | --- | --- | --- | --- | --- | --- | --- | --- |
| Site count | n = 2 | n = 2 | n = 5 | n = 2 | n = 5 | - | n = 7 | n = 3 |
| Age, year | 30; 80 | 44; 74 | 24; 45 | 44; 61 | 60; 141 | - | 40; 162 | 81; 96 |
| Mean height, m | 13; 20 | 16; 28 | 13; 18 | 9; 20 | 12; 21 | - | 10; 23 | 15; 20 |
| Mean diameter, cm | 12; 21 | 16; 28 | 12; 22 | 8; 21 | 12; 22 | - | 10; 25 | 17; 21 |
| Basal area, m$^2$ ha$^{-1}$ | 26; 36 | 30; 36 | 15; 23 | 22; 23 | 17; 48 | - | 18; 36 | 25; 42 |

At each study site, three subplots (Figure S1) for data collection were selected, with a minimum distance of 30 m, in an area representing typical ecosystem characteristics of the floristically defined site type, as classified under the local forest site type system (Bušs, 1981). The subplots were arranged along a transect positioned perpendicular to the closest ditch in drained areas and perpendicular to stand border in undrained areas. In the drained sites, the first subplot was located about 20 m from the nearest drainage ditch. In the undrained sites, the first subplot was located at least 20 m from the forest border.

Empirical data was gathered from January 2021 to December 2022 in Estonia and Latvia, and from July 2021 to June 2023 in Lithuania. The sites were visited monthly in Latvia and Lithuania, and biweekly in Estonia. The meteorological conditions during the study were typical for the region (Table S3).

**2.2 Respiration**

Measurements of forest floor respiration, which we will hereafter refer to as total respiration (Rtot), included both soil heterotrophic respiration and autotrophic dark respiration of ground vegetation (above and below-ground) as well as tree roots extending to the measurement locations. Gas samples were collected from manual closed static dark (opaque) chambers (PVC, volume 0.0655 $m^3$) as described in the literature (Hutchinson and Livingston, 1993) for subsequent laboratory analysis. Rtot monitoring included five to six monitoring locations, divided over 3 subplots, at each site. Ring-shaped chamber collars (area 0.196 $m^2$) were permanently installed in the soil at a depth of five cm at least one month before the first sampling to avoid the installation effect on fluxes. Collar locations reflected local diversity in vegetation and potential WTL gradient at each subplot and the encircled soil surface and vegetation were kept intact. Thus, the soil heterotrophic respiration component in Rtot includes $CO_2$ emissions caused by the decomposition of both fresh litter and soil organic matter.

Gas samples were collected by obtaining four air samples from a closed chamber into pre-evacuated (0.3 mbar) glass vials (100 $cm^3$). The air samples were taken from the chamber outlet, equipped with a valve attached in the sampling tube reaching approximately the centre of the airspace. The air within the chamber was not artificially mixed during sampling. Air sampling was done by first removing the residual air left from the sampling tube (to avoid potential impact on the concentration readout) by a syringe, and thereafter pre-evacuated glass vial was attached into the outlet. The first sample was taken immediately after attaching the chamber on the collar, and subsequent samples were taken at either 10 (Latvia) or 20 (Estonia and Lithuania) minute intervals over 30- or 60-minute monitoring periods, respectively (Butlers et al., 2022; Vigricas et al., 2024).

The gas samples were analysed using a Shimadzu GC-2015 gas chromatograph (Shimadzu USA Manufacturing, Inc., Canby, OR, USA) equipped with an electron capture detector (ECD). The uncertainty of the method used was estimated to be 20 ppm of $CO_2$ (Magnusson et al., 2017). Linear regression was applied to relate the $CO_2$ concentrations with the time elapsed since chamber closure for each measurement. Subsequently, the measurement data was screened to identify deviations from the recognized trend, considering the removal of measurements with identified errors. All measurements were discarded if the regression coefficient of determination ($R^2$) was less than 0.9 (p<0.01), except for cases where the difference between the highest and lowest measured $CO_2$ concentration in the chamber was less than the uncertainty of the method (specifically applicable during non-vegetation periods). Consequently, a small amount of data (<5%) was discarded.

The data that met the quality criteria were used to determine the slope coefficient of the linear regression, which was then used to calculate the instantaneous Rtot according to the ideal gas law equation (Fuss and Hueppi, 2024):

$$Rtot = \frac{M \times P \times V \times slope}{R \times T \times A \times 1000} \qquad (1)$$

where Rtot is the instantaneous total respiration, mg $CO_2$-C $m^2$ $h^{-1}$; M is the molar mass of $CO_2$-C, 12.01 g $mol^{-1}$; R is the universal gas constant, 8.314 $m^3$ Pa $K^{-1}$ $mol^{-1}$; P is the assumption of air pressure inside the chamber, 101.300 Pa; T is the air temperature in the chamber, K; V is the chamber volume, 0.0655 $m^3$; the slope is the $CO_2$ concentration change over time, ppm $h^{-1}$; and A is the collar area, 0.19625 $m^2$.

We also conducted measurements of heterotrophic respiration (Rhet) for comparison, as described in the Supplementary text.

**2.3 Environmental variables**

Manual WTL measurements were carried out using nylon-mesh-coated, perforated piezometer tubes (5 cm in diameter) installed down to a 140 cm depth in all subplots. Manual soil temperature measurements were done at depths of 5, 10, 20, and 40 cm in all subplots by Comet data logger (COMET SYSTEM, s.r.o., 756 61 Roznov pod Radhostem, Czech Republic) equipped with Pt1000 temperature probes. All manual measurements were carried out at the same time as the $CO_2$ flux

measurements. Continuous soil temperature measurements at depths of 10 and 40 cm were carried out at 30-minute interval in the centermost subplot (Maxim Integrated DS1922L2F, iButtonLink Technology, Whitewater, WI 53190 USA).

Once per month, soil water samples were collected from perforated tubes (7.5 cm in diameter) explicitly installed for water sampling for chemical analysis. Water chemical parameters including pH, electrical conductivity (EC), and concentrations of dissolved organic carbon (DOC), total nitrogen (N), and nitrate ($NO_3^-$), ammonium ($NH_4^+$) and phosphate ($PO_4^{3-}$) ions were determined. Soil samples were taken down to a depth of 75 cm (0-10; 10-20; 20-30; 30-40; 40-50; 50-75cm) at two locations in each subplot during the establishment of the study sites. Two separate sample sets were collected, one for the determination of bulk density, and another for ash content and chemical parameters (pH, concentrations of total carbon (TC), nitrogen (N), and $HNO_3$ extractable phosphorus (P), potassium (K), calcium (Ca), and magnesium (Mg)). The samples were collected with a volumetric 100 $cm^3$ cylinder (Cools and De Vos, 2010) at 10 cm intervals to a depth of 50 cm. Two additional samples were taken from soil depths of 50-75 and 75-100 cm with a soil auger. Soil samples collected for determination of bulk density were oven-dried (105 °C) and weighed, while soil samples for chemical analyses were prepared by air drying (≤40 °C), sieving and homogenizing (LVS ISO 11464:2006). Organic carbon (Corg) content was calculated by multiplying soil organic matter content derived from the ash content measurement result by factor 0.5, thus assuming that organic matter is 50% Corg (Pribyl, 2010). All soil and water analyses were done in an ISO 17025-certified laboratory using ISO standard methods (Table S4).

**2.4 Litter input**

Annual litter inputs to be used in $CO_2$ balance estimation were either measured (foliar litter) or estimated based on biomass components measured (ground vegetation, fine roots).

Foliar fine litter (fLF) was collected with conical litter traps (area 0.5 $m^2$) set one meter above ground (Latvia, Lithuania), or with square mesh frames (0.5 x 0.5 m) placed on the ground (Estonia). In each study site, five replicate litter traps were placed in the centermost subplot of the transect. It included all fine fractions of litter, such as needles, leaves, and branches with a diameter up to 1 cm and a length up to 10 cm. Branches with larger dimensions were collected from coarse woody litter (cLF) traps (square mesh frames, 0.5 x 0.5 m) placed on the ground. The litter samples were collected from the traps once every four weeks. Due to the heterogenous nature and large dimensions of cLF, respective decomposition emissions could not be representatively included in Rtot measurements. Therefore, we considered only fLF as the litter input source, and the cLF results are presented only as indicative of the site conditions.

Ground vegetation (GV) aboveground (aGV) and belowground (bGV) biomass samples were collected at the end of the growing season (August) in 2021 in five replicates per subplot, from square sampling locations with an area of 0.0625 $cm^2$. Aboveground biomass was separated into herbaceous (aGV) species and dwarf shrubs. bGV was collected from the top soil layer, extending down to 20 - 30 cm from the same location and area as aGV. Herbaceous ground vegetation roots (bGV) were separated from tree and shrub roots by wet sieving based on root morphological properties. We assumed that the measured herbaceous ground vegetation aboveground (aGV) and belowground (bGV) biomasses were equal to annual ground vegetation litter inputs. The contribution of shrub litter was thus not included but was assumed to be minor (see Table S5).

Moss biomass was measured by collecting samples from 0.01 $m^2$ square areas, with four replicates per subplot, and visually removing any dead parts. Collections were conducted concurrently with moss production sampling. Moss production (MP) samples in four replicates per subplot were collected by anchoring a square mesh (0.01 $m^2$) on the moss at the end of the growing season and harvesting the moss biomass grown through the mesh by the end of the next growing season. Including moss litter in the $CO_2$ balance estimation by aligning $CO_2$ influx and efflux from areas with and without moss cover, without increasing uncertainty, would however have required doubling the number of spatial replicates in chamber measurements. Therefore, we did not consider moss as a litter source, which causes underestimation of litter inputs to a varying extent. The moss results are presented only as indicative of site conditions (see Table S5).

Tree fine-root production (FRP) was estimated with the ingrowth core method (Laiho et al., 2014; Bhuiyan et al. 2017). Five replicate ingrowth cores (cylindrical mesh bags with diameter 2.5 cm, mesh size 2 mm) per subplot, filled with soil collected from the subplot, were installed in autumn or spring and removed after two growing seasons. In the laboratory, the biomass of the ingrown fine roots was determined after wet sieving, and ground vegetation roots were separated from tree roots by morphological properties. We assumed that tree fine-root biomass was essentially not changing over the study years, and thus we could assume that FRP equalled litter production. Since the ingrowth cores were removed from the soil after two growing seasons, the FRP estimate was calculated by dividing the fine-root biomass in the cores by two (Bhuiyan et al., 2017).

All litter and biomass samples were oven-dried (70 °C), weighed and milled before further analysis. Chemical analyses were performed according to ISO standard methods (Table S4). To estimate the annual $CO_2$ influx to be used in the $CO_2$ balance estimation (section 1.5), the estimated annual litter inputs on a unit area were transformed to $CO_2$ influx by using C content values measured for each component (Table 2).

**Table 2: Mean C and N content (% of dry matter) in foliar litter from trees and biomass components used for estimating other litter inputs.** The values are means ± standard deviation including both drained and undrained sites. Abbreviations: aGV and bGV – above- and belowground biomass of herbaceous vegetation, FR – tree fine roots, M – moss, fLF – foliar fine litter, cLF – coarse woody litter.

| Element | aGV | bGV | FR | M | fLF | cLF |
|---------|------|------|------|------|------|------|
| C | 49.34±2.45 | 50.95±2.02 | 51.21±5.16 | 48.38±2.13 | 52.50±0.25 | 53.88±0.67 |
| N | 2.18±0.64 | 1.53±0.43 | 1.47±0.44 | 1.10+0.75 | 1.30±0.41 | 1.04±0.20 |

**2.5 Estimation of annual soil $CO_2$ balance**

We estimated the annual soil $CO_2$ balance of the sites by combining annualized $CO_2$ influx (section 1.4) and efflux data. In exceptional cases where the $CO_2$ influx from specific source was not estimated in certain countries, such as bGV and FRP in EE and FRP in some sites in LT and LV (Table S10), we assumed the $CO_2$ influx to be equivalent to the average results from sites with the same drainage status in the other countries. While we directly measured Rhet, we utilized the estimated Rhet derived from Rtot as the efflux value (marked from here onwards as Rhet`). Such an approach was necessary because, at all sites, our Rhet values were significantly higher (by mean 5.8±3.1 t $CO_2$-C ha$^{-1}$ year$^{-1}$) than Rtot, which would be logically impossible if the conditions in the measurement locations were the same. This pattern was probably mostly due to the high and variable $CO_2$ efflux from roots killed by the trenching, noted also in other studies (e.g., Hermans et al., 2022). This discrepancy could not be remedied with the root data at hand (for more details, please refer to the Supplementary text and Discussion section).

To estimate annual $CO_2$ efflux, at first, site-specific relationships between instantaneous Rtot fluxes and soil temperatures were established. To identify the best approach for to express the relationship, we compared the suitability of exponential regression on untransformed data and linear regression on logarithmically or Box-Cox transformed data. The performance of these models for flux data interpolation based on continuous soil temperature data was evaluated by using the root mean square error (RMSE) of prediction. It was found that the Box-Cox approach achieved the best conformity of flux data to a normal distribution and provided the best fit for the models. Previous studies also indicate that this method effectively addresses the typical underestimation of fluxes caused by their nonlinearity (Box and Cox, 1964; Liaw et al., 2021; Wutzler et al., 2020). Interpolation was performed by evaluating the relationship between Rtot and soil temperature (at 10-cm depth) measured at each study site and constructing site-specific linear regression equations (Table S6) after applying Box-Cox transformation to the flux data for normalization (Box and Cox, 1964). This approach, compared to alternative methods, more successfully (lower RMSE of prediction) accounted for the exponential nature of the relationship between flux and temperature, and prevented underestimation (Liaw et al., 2021; Wutzler et al., 2020). Hourly Rtot estimates were then formed by using the hourly recorded soil temperatures (logger data) for each study site. Consecutively, site-specific annual Rtot estimate was calculated by summing the interpolated hourly emission estimates of the year.

We derived site-specific annual Rhet` from estimated annual Rtot empirically. To avoid potential bias introduced by using a single fixed factor, we instead used Equation (2), accounting for the observed pattern of decreasing Rhet proportion as soil surface respiration (Rs) increases (Bond-Lamberty et al., 2004; Subke et al., 2006). The equation characterizes the relationship between Rs and Rhet, and was created using results of previous studies (Jian et al., 2021) in the boreal zone (Figure S2). We assumed Rtot is equal to Rs, i.e., that aboveground autotrophic respiration has a minor role in Rtot (Hermans et al., 2022; Munir et al., 2017), and applied the equation to annual Rtot directly.

$$Rhet` = -0.70 + 0.78 \times Rtot \qquad (2)$$

Neglecting aboveground autotrophic respiration leads to a minor overestimation of Rhet.

We will use the + sign to denote positive soil $CO_2$ balance (soil $CO_2$ sink), and – sign to denote net loss of $CO_2$ from soil to the atmosphere.

**2.6 Statistical analysis**

Statistical analyses were performed and figures prepared using the software R version 4.3.1 (packages 'MASS', 'stats', 'nlme', 'Hmisc', 'lmerTest', 'lme4', 'vegan', 'pls', and 'caret'), using p=0.05 as the limit for statistical significance. The compliance of the data with the normal distribution was checked with the Shapiro-Wilk normality test and visually by density and quantile-quantile (Q-Q) plots. To calculate the uncertainty of the study results when combining multiple data sources, we used the root sum of the squares method to aggregate the individual uncertainties (95 % confidence interval). Therefore, for the $CO_2$ influx, for instance, we combined the uncertainties from various influx (fLF, GV, FRP) sources. The uncertainty of Equation (2) used for the calculation of Rhet` was expressed as the root mean square error (RMSE) of the corresponding regression model. The soil $CO_2$ balances were calculated by summing the $CO_2$ influx and efflux for individual sites. The uncertainties of the averaged $CO_2$ balances, categorized by drainage status, country, site type, or dominant tree species, are expressed as standard error. Figures were prepared by using packages 'ggplot2', 'corplot', 'ggbiplot'.

Correlations between the sample groups were expressed with the Pearson correlation coefficient (r). We compared differences between two sample groups using pairwise Wilcoxon rank sum tests with continuity correction and adjusted the p-values using Bonferroni correction. The method was used to compare soil parameters and instantaneous or annualized Rtot data segregated by site type, drainage status, country or dominant tree species. In the same way, soil characteristics were compared between drained and undrained sites. Multivariate testing of flux and impacting factor relationships involved assessing the significance of these factors on the relationship between soil temperature and Rtot using mixed-effects linear models. As flux impacting factors, we considered country, dominant tree species, drainage status, WTL, and a 30 cm WTL threshold distinguishing shallow and deep drainage in the IPCC guidelines. In this analysis, Box-Cox transformed flux data was fitted to linear models using the study site as a random effect. In addition, multivariate relationships were observed through Principal Component Analysis (PCA) to visualize covariation and seek observational confluences with the results of other analyses. To assess the contribution of influencing factors on the soil $CO_2$ balance, Redundancy Analysis (RDA) and Partial Least Squares Regression (PLSR) were conducted.

**3 Results**

**3.1 Soil and soil water characteristics**

The organic layer depth in the drained sites ranged from 27 to 212 cm (mean 81±47 cm) and in undrained sites from 100 to 230 cm (mean 167±49 cm). Soil bulk density (0-30 cm depth) in the drained sites (mean 314±215 kg m$^{-3}$) was characterized by both higher variation and higher mean density (p=0.003) compared to undrained sites (mean 168±32 kg m$^{-3}$) (Table S7). Soil drainage status had no impact on Corg content in the 0-30 cm soil layer (p=0.11, total mean 416±130 g kg$^{-1}$). However,

drained soils had a higher mean C:N ratio (22±7; p=0.01) than the undrained soils (17±3). A trend could be observed that undrained soils had higher nutrient concentrations and higher pH than the drained soils. Soil analysis confirmed that site types classified based on ground vegetation composition served as an indicator of nutrient availability (Figure 2). In drained Oxalis sites, mean macronutrient concentrations in the 0–30 cm soil layer was higher compared to Myrtillus sites. Notably, N, Mg, Ca, and P showed statistically significant differences ($p < 0.05$). Additionally, soil pH values in the Oxalis sites were, on average, 1.89 units higher ($p = 0.018$).

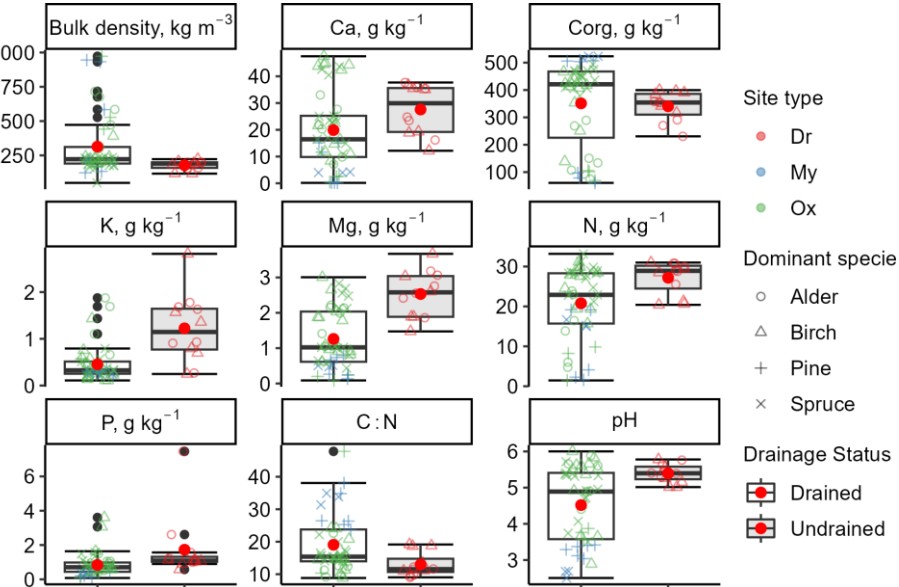

**Figure 2: Variation of soil chemical and physical properties at soil depth 0-30 cm.** The clear box represents the drained, and the grey-shaded box the undrained sites. Data points represent individual subplots. The bottom and top edges of the box represent the 25th and 75th percentiles, summarizing the interquartile range (IQR). The whiskers extend to the smallest and largest values within 1.5 × IQR from the 25th and 75th percentiles, respectively. Black dots mark outliers. A red dot and a solid horizontal line in the box indicate mean and median values, respectively. Corg – organic carbon; N – total nitrogen. Site types: Dr - *Dryopterioso–caricosa*; Ox - *Oxalidosa turf. mel.*; My - *Myrtillosa turf. mel*.

The range of mean WTL over the study period was from −23 to −112 cm (mean −60±25 cm) in the drained sites and from −7 to −17 cm (mean −13±4 cm) in the undrained sites, respectively. In the undrained sites, the WTL was mainly rather high (see interquartile range in Figure 3) and had comparably smaller variation (mean standard deviation 16 cm) than in the drained sites (mean standard deviation 23 cm); however, in all sites except LTC108, WTLs below 30 cm were also observed (Figure 4). In the undrained sites, the range of min-max WTL was from 3±3 cm to −63±27 cm, while the WTL in drained sites had a greater absolute variation ranging from −14±19 cm to −104±28 cm.

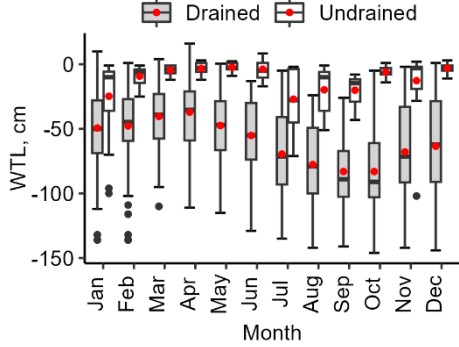

**Figure 3: Yearly variation of soil water-table level (WTL) in the study sites.** The edges of the box represent the 25th and 75th percentiles, encapsulating the interquartile range (IQR). The whiskers extend to the smallest and largest values within 1.5 * IQR from the 25th and 75th percentiles, respectively. Black dots mark outliers. A red dot and a solid horizontal line indicate the average values of the date represented – mean and median, respectively.

The concentrations of all measured chemical parameters in the soil water, except for $NH_4^+$, were, on average, higher in the drained sites (Figure S3). However, the results were highly variable (Table S8).

### 3.2 Instantaneous total respiration

In the drained sites, the mean instantaneous Rtot varied from 48 to 125 mg $CO_2$-C $m^{-2}$ $h^{-1}$, and in the undrained sites from 38 to 80 mg $CO_2$-C $m^{-2}$ $h^{-1}$ (Figure 4). In all sites combined, during the summer months (June, July, August), the interquartile range of Rtot varied from 111 to 198 mg $CO_2$-C $m^{-2}$ $h^{-1}$ with a mean of 160±78 mg $CO_2$-C $m^{-2}$ $h^{-1}$. In contrast, during the winter (December, January, February), it ranged from 8 to 24 mg $CO_2$-C $m^{-2}$ $h^{-1}$ with a mean of 17±14 mg $CO_2$-C $m^{-2}$ $h^{-1}$. The relative standard deviations of the instantaneous Rtot in drained (90±9%) and undrained (106±29%) sites were comparable. Although the study sites represented a broad soil WTL gradient, no significant impact of the site mean WTL on the mean instantaneous Rtot emission was observed (r=0.16, p>0.05). Furthermore, no significant correlations were found between instantaneous Rtot and soil water parameters.

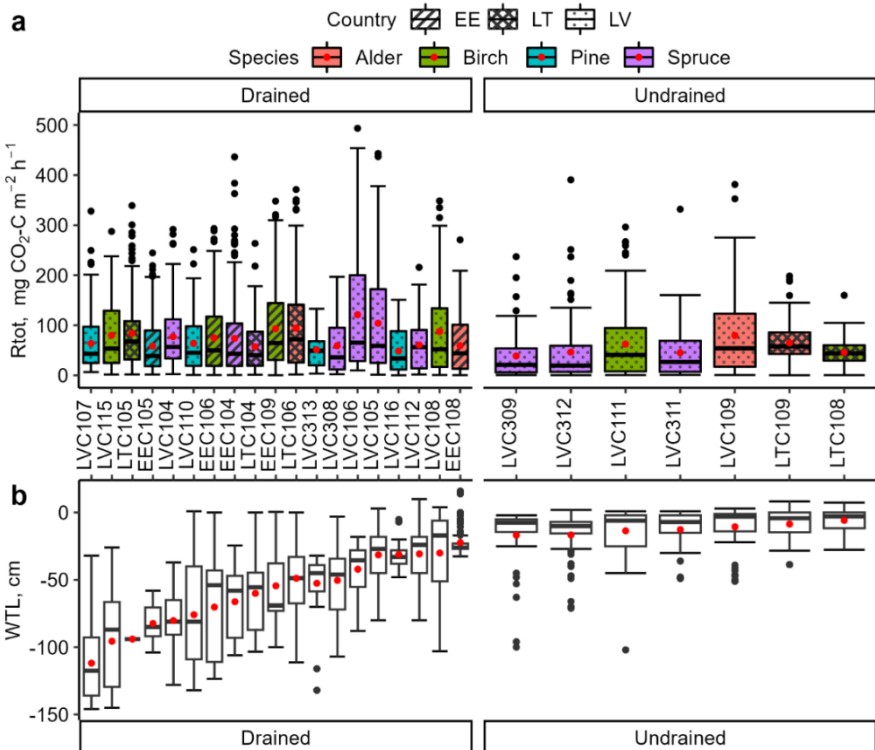

**Figure 4: Variation of instantaneous total respiration (Rtot, a) and soil water-table level (WTL, b) in the study sites.** The bottom and top edges of the box represent the 25th and 75th percentiles, summarizing the interquartile range (IQR). The whiskers extend to the smallest and largest values within 1.5 × IQR from the 25th and 75th percentiles, respectively. Black dots mark outliers. A red dot and a solid horizontal line in the box indicate mean and median values, respectively.

The instantaneous Rtot in drained sites (mean 76±3 mg $CO_2$-C $m^{-2}$ $h^{-1}$) was overall higher (p<0.05) than Rtot in undrained sites (mean 56±5 mg $CO_2$-C $m^{-2}$ $h^{-1}$) (Figure 5a). . Rtot was the lowest at undrained sites dominated by spruce and the highest at drained sites dominated by birch (Figure 5b). Rtot was significantly different (p<0.05) between coniferous forest sites with different dominant tree species and/or soil moisture regime, with Rtot ranging from mean 42±7 mg $CO_2$-C $m^{-2}$ $h^{-1}$ in undrained spruce forests to 59±4 and 81±6 mg $CO_2$-C $m^{-2}$ $h^{-1}$ in drained pine and spruce forests, respectively. In deciduous stands, the moisture regime and dominant tree species had less impact on the mean flux; Rtot was higher (p<0.05) in drained birch stands (mean 84±5 mg $CO_2$-C $m^{-2}$ $h^{-1}$) than in undrained birch stands (56±8 mg $CO_2$-C $m^{-2}$ $h^{-1}$), while in alder stands the mean Rtot was similar regardless of the soil moisture regime (total average 67±9 mg $CO_2$-C $m^{-2}$ $h^{-1}$) (Figure 5b). In drained coniferous

and deciduous sites, on average, the mean Rtot was similar, but in undrained sites, emissions in deciduous forests were about
40% higher. Mean Rtot in sites with the same drainage status did not differ (p>0.05) between countries (Figure **S4**, g).

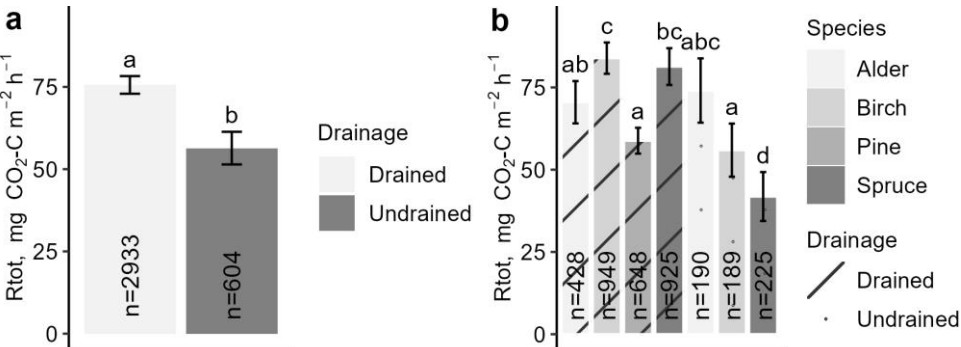

**Figure 5: Mean instantaneous total respiration (Rtot) throughout the study period, categorized by drainage status (a) or dominant**
**tree species and drainage status (b).** Error bars indicate confidence interval. A shared letter indicates that differences are not significant.
The impact of WTL is reflected in the mean Rtot, which was $87\pm3$ mg $CO_2$-C $m^{-2}$ $h^{-1}$ when WTL was below 30 cm and
$57\pm3$ mg $CO_2$-C $m^{-2}$ $h^{-1}$ when WTL was closer to soil surface. However, when evaluating the effect of WTL on Rtot variation
in mixed-effects models predicting Rtot based on soil temperature, WTL was found to have an insignificant impact on Rtot
variation. Similarly, the contribution of country and dominant tree species to Rtot prediction was marginal (Table S9). The
inclusion of dominant species provided minimal model improvement ($\Delta$AIC = +5, $\Delta$logLik = 0), while country effects captured
some additional variability in Rtot. However, the increase in $R^2$ due to country variables was minor (from 0.77 to 0.78),
indicating limited explanatory power.
**3.3 Annual total respiration**
Soil temperature at 10 cm depth (Figure S5) was used in constructing Rtot prediction models and for emission interpolation
needed for annualizing Rtot. The 10 cm depth was chosen because it showed the strongest correlation between instantaneous
Rtot and soil temperatures measured at different depths, with a mean Pearson correlation coefficient (r) of $0.86\pm0.04$ across
the study sites. For the other soil depths (5, 20, 30, 40 cm), r ranged from $0.71\pm0.07$ to $0.79\pm0.05$. Linear models developed
using Box-Cox transformed data provided the best Rtot prediction power. A lambda value of 0.3411 was used for all data
transformations, as individual data transformations for each site resulted in comparatively less successful data normalization.
With this approach, the RMSE of instantaneous Rtot predictions for individual sites decreased by an average of $16\pm14\%$,
compared to linear models with logarithmically transformed data or non-linear models with untransformed data (Table S6).
Annualized Rtot indicated similar mutual relationships among the study site dominant tree species and drainage status
categories as the instantaneous Rtot. Pooled estimated annual emissions from drained sites (overall mean $6.41\pm0.49$
t $CO_2$-C $ha^{-1}$ $year^{-1}$) and undrained sites (overall mean $4.75\pm0.68$ t $CO_2$-C $ha^{-1}$ $year^{-1}$) differed significantly (Figure 6, a).
However, when the data were further stratified by country, the differences were no longer statistically significant due to wider
confidence intervals (Figure S4, h).

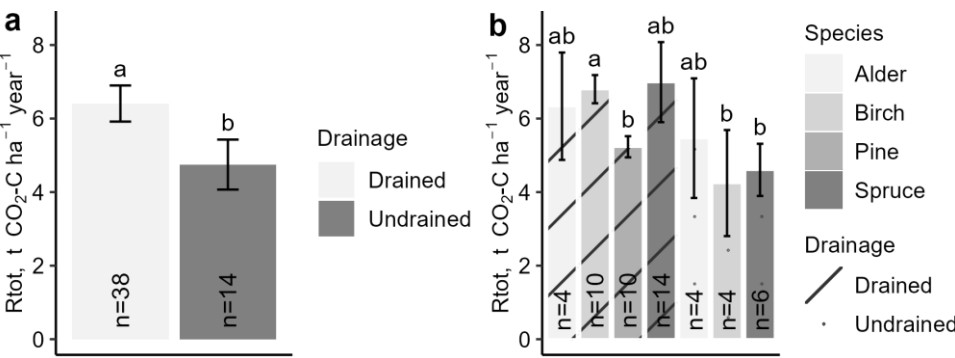

**Figure 6: Annualized total respiration (Rtot) in study sites stratified by drainage status (a) or dominant tree species (b).** Error bars indicate confidence interval. A shared letter indicates that differences are not significant.

When categorizing data according to drainage status and dominant tree species, fewer differences were found in the annualized Rtot than in the instantaneous Rtot (Figure 6, b). For instance, among the drained sites, the lowest mean annual Rtot was estimated for pine forests ($5.23 \pm 0.29$ t $CO_2$-C ha$^{-1}$ year$^{-1}$), while in spruce, birch, and alder forests, the means were similar ($p > 0.05$) ($6.71 \pm 0.31$ t $CO_2$-C ha$^{-1}$ year$^{-1}$). Emissions from undrained soils in alder, birch, and spruce forests were similar to each other and lower than from drained sites, ranging from $4.6 \pm 0.71$ in spruce forests to $5.47 \pm 1.63$ t $CO_2$-C ha$^{-1}$ year$^{-1}$ in alder forests (overall mean $4.86 \pm 0.71$).

The correlation between Rtot and WTL was low; however, a drainage status (drainage ditch presence) impact on Rtot is indicated by the PCA results, where undrained sites tend to have more similar characteristics while drained sites show greater diversity concerning Rtot. However, clear covariation of dominant tree species and Rtot are not recognized by PCA (Figure S6 and Figure S7). When comparing the chemical and physical properties of different soil layers with the estimated annual Rtot, as well as the measured mean Rhet, the mean measured Rhet consistently showed a higher correlation with evaluated soil parameters (Figure S8). The only exception is Corg, for which no correlation was observed between Corg in different soil layers and neither Rtot or Rhet ($r$ around $-0.1$). Excluding Corg, the other soil chemical parameters generally had a low to moderate correlation (mean $r = 0.4$) with respiration. The highest correlation was with pH, K, Mg, and P (mean $r = 0.5 \pm 0.07$, $p < 0.05$), and it was consistent across all evaluated soil layers, while the correlation with BD (mean $r = -0.2$, $p > 0.05$) tends to increase with deeper soil layers reaching the highest correlation ($r = -0.3$) in layer 20-30 cm. In addition, a higher C:N ratio is associated with lower $CO_2$ emissions (mean $r = -0.4$, $p < 0.05$).

### 3.4 Annual litter inputs

The estimated mean litter inputs at the subplot level were mostly similar between drained and undrained sites (Table S10 and S11), typically differing by less than 20%. Only fLF and FRP tended to be considerably higher in the drained sites, FRP on average even more than twice as high. Compared to undrained sites, bGV in drained sites was about 20% higher on average, while aGV was about 20% lower on average (Figure 7). However, regardless of the soil drainage status, the proportion of aGV in the total GV biomass was $54 \pm 18\%$ (Table 3).

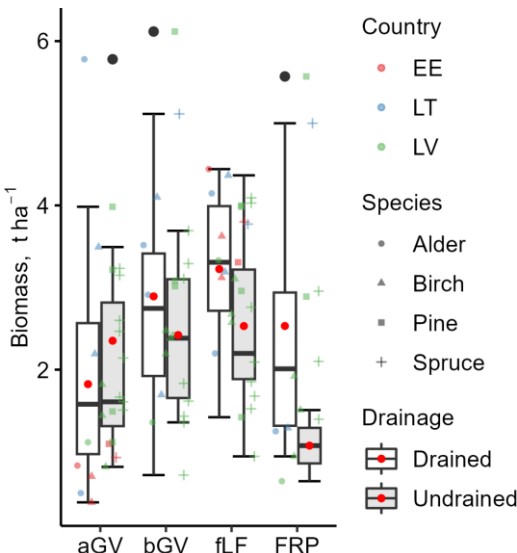

399

400

**Figure 7: Variation in the biomass components used as litter input estimates.** Abbreviations: aGV and bGV – above- and belowground biomass of herbaceous vegetation, fLF – fine foliar litter (needles, leaves, fine woody litter), FRP – tree fine root production. The bottom and top edges of the box represent the 25th and 75th percentiles, summarizing the interquartile range (IQR). The whiskers extend to the smallest and largest values within 1.5 × IQR from the 25th and 75th percentiles, respectively. Black dots mark outliers. A red dot and a solid horizontal line in the box indicate mean and median values, respectively.

**Table 3: Biomass (mean±CI, t dm. ha$^{-1}$) components used as litter input estimates, stratified by drainage status.** Abbreviations: aGV and bGV – above- and belowground biomass of herbaceous vegetation; respectively; FRP – tree fine root production; fLF – fine foliar litter (needles, leaves, fine woody litter).

| Category | Drained | Undrained |
|---|---|---|
| aGV | 1.82±0.52 | 2.35±1.61 |
| bGV | 2.89±0.85 | 2.42±0.84 |
| FRP | 2.53±0.77 | 1.08±0.57 |
| fLF | 3.22±0.44 | 2.53±1.06 |

409

Both bGV (r=|0.6|) and FRP (r=|0.7|) had a significant negative correlation with soil pH but a positive with the C:N ratio in soil layer 0-30 cm. Additionally, FRP had a significant negative correlation (r=|0.7|) with the contents of N, Ca, and Mg in the soil. No explanatory factors for aGV could be identified. Moderate correlation (r=0.5, p<0.05) was found between stand age and fLF.

**3.5 Annual soil carbon balance**

The estimated Rhet` (Table S10) proportion of Rtot varied between 54 and 71% (mean 65%). Consequently, the estimated annual gross C losses from drained soils in the form of Rhet` emissions ranged from 2.36 to 7.49 t $CO_2$-C ha$^{-1}$ year$^{-1}$ (mean 4.30±1.20), while for undrained soils the range was from 1.63 to 4.68 t $CO_2$-C ha$^{-1}$ year$^{-1}$ (mean 3.00±0.99). According to the RMSE of the model (Equation 2, Figure S2) prediction, the Rtot to Rhet` calculation introduced an uncertainty of approximately 0.32 t $CO_2$-C ha$^{-1}$ year$^{-1}$. In drained and undrained sites, the total estimated $CO_2$ influx ranged from 3.81 to 7.03 t $CO_2$-C ha$^{-1}$ year$^{-1}$ (mean 5.20±0.91) and 2.89 to 5.98 t $CO_2$-C ha$^{-1}$ year$^{-1}$ (mean 4.19±1.10), respectively (Figure 8). The uncertainties (relative CI) in $CO_2$ efflux and influx in both drained and undrained soils were relatively uniform, with a mean uncertainty of 36±14% for the estimated individual annual $CO_2$ fluxes. The largest source of uncertainty was $CO_2$ influx in undrained soils (49±13%), while the uncertainty of individual $CO_2$ fluxes in drained sites averaged 27±6% (Figure 8).

424

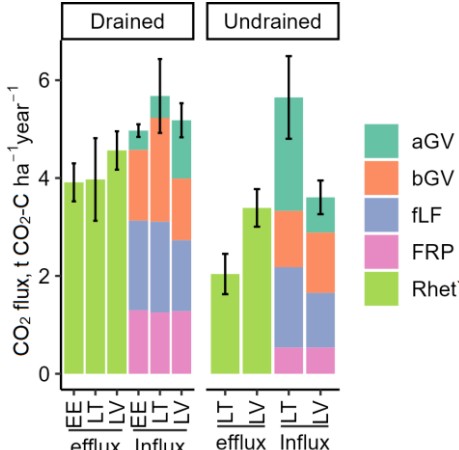

**Figure 8: Components of the estimated soil CO₂ balance (sum±combined CI).** Efflux is soil heterotrophic respiration (Rhet`) calculated from Rtot, and influx is the estimated litter input; both are expressed as C. Thus, efflux indicates soil C losses and influx soil C gains. Abbreviations: aGV and bGV – above- and belowground biomass of herbaceous vegetation; respectively; fLF – fine foliar litter; FRP – tree fine root production.

The mean soil $CO_2$ balance during the study period was $+1.06\pm0.45$ and $+1.27\pm0.73$ t $CO_2$-C ha$^{-1}$ year$^{-1}$ for the drained and undrained sites, respectively (Figure 9, a). These results indicate that long-term drainage reduced soil $CO_2$ sequestration capacity by an average of 0.20 t $CO_2$-C ha$^{-1}$ year$^{-1}$.

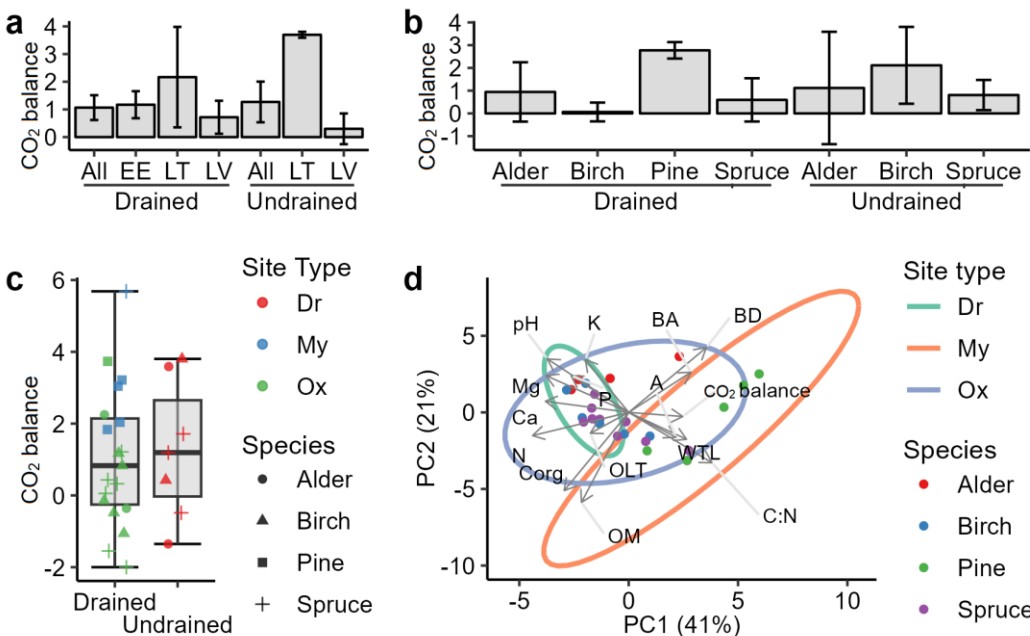

**Figure 9: Soil CO₂ balance (a, b, c: mean t CO₂-C ha⁻¹ year⁻¹±SE) and impacting factors (d: PCA biplot). Positive values indicate soil CO₂ sink, i.e., CO₂ removals from the atmosphere.** Abbreviations: A – stand age; BA – basal area, BD – bulk density; C:N – ratio between organic carbon and nitrogen in soil; OLT – soil organic layer thickness; WTL – water table level; pH – soil pH value; K, Ca, Mg, P, OM, Corg, N represent the content of potassium, calcium, magnesium, phosphorus, organic matter, organic carbon and nitrogen in the top 0-30 cm layer of soil, respectively. Site types: Dr - Dryopterioso–caricosa (undrained sites); Ox - Oxalidosa turf. mel. (drained); My - Myrtillosa turf. mel. (drained).

The soil $CO_2$ balance stratified by dominant tree species (Figure 9, b) indicates that the drained sites were skewed towards $CO_2$ removals due to significantly higher $CO_2$ removals in pine stands. Pine sites (n=5) showed high $CO_2$ removals with low uncertainty (mean $+2.77\pm0.36$ t $CO_2$-C ha$^{-1}$ year$^{-1}$), in contrast to the other drained sites (n=19), where the mean soil $CO_2$ balance was estimated at $+0.45\pm0.50$ t $CO_2$-C ha$^{-1}$ year$^{-1}$, suggesting that soils in drained alder, birch and spruce sites were in near $CO_2$ equilibrium during the study period. The soil $CO_2$ removals identified for undrained birch and spruce stands

(+1.33±0.72 t $CO_2$-C ha$^{-1}$ year$^{-1}$, n=5) were consistent, while in alder stands, the $CO_2$ balance of +1.12±2.47 t $CO_2$-C ha$^{-1}$
year$^{-1}$ was highly uncertain due to the number of sites being just two.
A trend of higher soil $CO_2$ removals was observed in site types associated with relatively lower nutrient availability (Figure 9,
c). In Oxalis sites, the mean soil $CO_2$ balance was +0.32±0.40 t $CO_2$-C ha$^{-1}$ year$^{-1}$, while in Myrtillus sites, it was
+3.16±0.69 t $CO_2$-C ha$^{-1}$ year$^{-1}$. Thus, drained nutrient-richer soils were approximately at $CO_2$ equilibrium, whereas
comparably nutrient-poorer soils acted as a $CO_2$ sink. The observed tendency is supported by PCA (Figure 9, d) which indicates
that higher soil $CO_2$ removals are associated with lower nutrient concentrations and pH levels. In addition, the PCA reveals
that the risk of soil $CO_2$ source is reduced in stands with higher basal area and age, as well as with a higher soil C:N ratio,
which in our study likely reflects the variability in peat quality and decomposability under different vegetation types.
According to correlation analysis, soil parameters such as pH, C:N ratio, N, and P showed the strongest correlations with the
soil $CO_2$ balance (Figure 10). Basal area was the tree stand characteristic with the strongest correlation with $CO_2$ balance,
while, among the $CO_2$ flux components, bGV demonstrated the most consistent role in $CO_2$ balance. No meaningful
relationship was identified between soil $CO_2$ balance and soil organic matter or C content, nor with the depth of the WTL or
organic layer thickness.

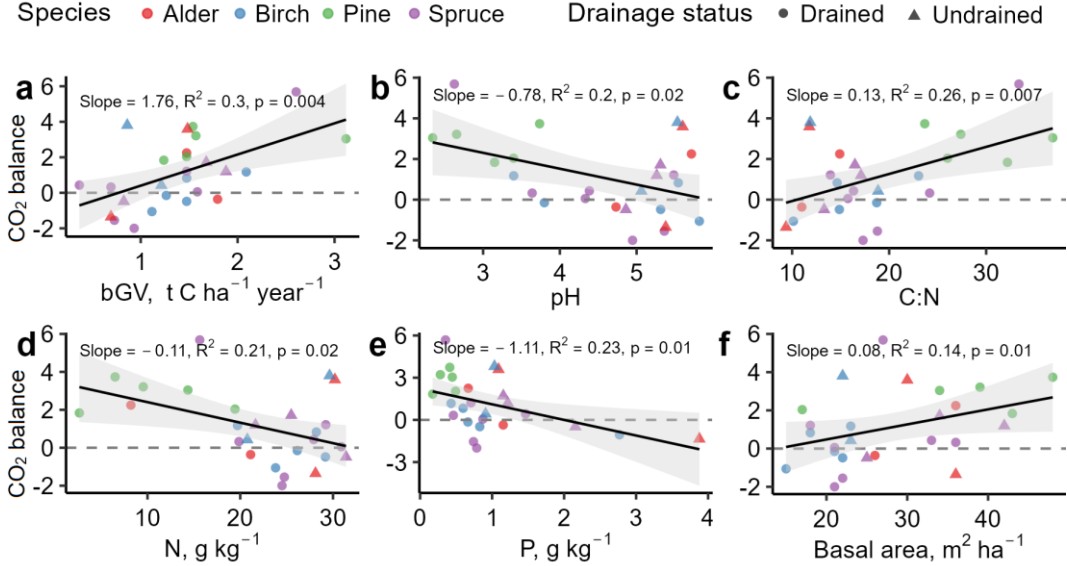


**Figure 10: Relationships between soil $CO_2$ balance (t $CO_2$-C ha$^{-1}$ year$^{-1}$) and impacting factors.** Positive values indicate soil $CO_2$
removals. The figures include impacting factors showing the highest correlations found in the study dataset. Data points represent individual
study sites.
Soil nutrient availability as $CO_2$ balance affecting factor is confirmed by RDA and PLSRS models. RDA and PLSRS models
(p<0.05) explained 78% and 70% of the soil $CO_2$ balance variance, respectively. After excluding variables introducing
multicollinearity, the models included WTL, organic layer thickness, pH, N, K, Ca, Mg, P, organic matter content of the soil,
stand age, and basal area. The variables pH, K, and Mg showed significant contributions (p < 0.05) in the RDA model
explaining the variation of soil $CO_2$ balance. The variable N was not significant (p = 0.087) but was close to the threshold of
significance. PLSR model indicated that pH, K, Mg and N explained 45% of the $CO_2$ balance variance; however, the VIP
values for all potential explanatory variables were below 0.4, suggesting limited predictive power for $CO_2$ balance with the
current dataset.

## 4 Discussion

### 4.1 Soil CO₂ balance

The soil $CO_2$ balance of the studied drained and undrained organic forest soils fluctuated around equilibrium, demonstrating both $CO_2$ sink and source dynamics. The reason for the uncertain $CO_2$ balance in undrained alder stands could not be determined, as site characteristics were consistent with the patterns observed in other undrained sites showing soil $CO_2$ removals. However, we identified soil properties as the likely reasons for why the soil in drained pine stands showed $CO_2$ removals, in contrast to the C-neutral soils observed at other drained sites.

Although observing a $CO_2$ sink in drained nutrient-rich soils may seem somewhat unexpected, given that these soils have generally been estimated to on average act as net $CO_2$ sources in both boreal and temperate zones (Jauhiainen et al. 2023), it is not entirely novel. Both soil $CO_2$ sinks and sources have also been observed in earlier studies under a wide range of site conditions (e.g., Ojanen et al. 2013; Minkkinen et al., 2018; Bjarnadottir et al., 2021; Hermans et al., 2022). Many of the soil $CO_2$ sink sites may be classified as nutrient-poor, but not all (e.g., Ojanen et al. 2013). In soil inventory studies carried out in Latvia, the soil C stocks of forestry-drained peatlands were found to be stable in all but the most nutrient-rich soil conditions, under which the C stock was reduced in the long term (Dubra et al., 2023; Lazdiņš et al., 2024). However, interpretation of soil $CO_2$ balances solely based on the nutrient status of a forest site should be approached with caution, as it is typically derived from indirect indicators such as vegetation or stand productivity, rather than a quantitative assessment of soil nutrient concentrations. Consequently, sites with variable conditions may be classified under a given nutrient status category. Similarly, drainage status does not guarantee specific WTL levels (Figure 4). This may be the reason for varying findings on soil $CO_2$ balances across studies that allegedly target the same soil type and drainage status, i.e. the category of drained, nutrient-rich soils may be too broad, encompassing varying nutrient and moisture regimes, which prevents the expectation of similar $CO_2$ balances, especially in different climates. In our study, this aspect was evident, as the soil $CO_2$ balance, regardless of drainage status, appeared to be influenced by the composition and relative proportions of the dominant tree species in combination with variation in soil nutrient conditions and organic matter quality, as indicated by the C:N ratio, across the study sites.

The soils at our study sites represented a wide range, from highly mineralized soils close to the threshold of organic soil definition (Hiraishi et al., 2013) to deep peat (Figure 2). Similarly, the WTLs varied widely, with sites ranging from an average WTL close to the soil surface to depths exceeding one meter (Figure 4). However, no meaningful relationship was identified between WTL variation and soil $CO_2$ efflux, nor between mean WTL and soil $CO_2$ balance. Also, soil C and organic matter contents, or the depth of the organic layer were poor predictors of soil $CO_2$ balance. Probably the importance of these factors decreases over time since initial disturbance by drainage system implementation. Rhet has been found to decrease over time following drainage (Qiu et al., 2021), except for the initially wettest, waterlogged hollow surfaces (Munir et al., 2017). There are few long-term monitoring or chronosequence studies, but those suggest that the most intensive period of soil C loss following drainage is the first decade/decades (Hargreaves et al. 2003; Vanguelova et al., 2019). During that period, also peat subsidence following drainage is highest (Lukkala 1949), and labile substances in peat that due to lowering WTL and peat subsidence becomes exposed to oxic decomposition are largely lost, leaving more decomposition-resistant substrates behind (e.g., Jayasekara et al., 2025). Simultaneously, major changes in litter inputs and their decomposability take place (Straková et al., 2010, 2012). Considering the long period since drainage of our sites, dating back to around a century ago and potentially even to the mid-19th century (Zālītis, 2012), the initially high soil C loss due to drainage-induced increase in gross soil $CO_2$ emissions has likely been offset by enhanced biomass growth and the resulting increased litter inputs (Hommeltenberg et al., 2014). However, a comparison of soil $CO_2$ balances between drained and undrained sites still shows a negative impact of historical drainage (Table S11). To more accurately assess the impact of drainage on soil $CO_2$ balance and its evolution in time, long-term studies would be required.

Soil nutrient conditions explained the observed $CO_2$ balances better than WTL and soil C characteristics did. In our study, drained nutrient-rich soils were represented by Oxalis (*Oxalidosa turf. mel.*) and Myrtillus (*Myrtillosa turf. mel.*) site types. All our drained birch and alder sites and most of drained spruce sites belonged to Oxalis site type, but most of the pine sites belonged to the Myrtillus site type. While the soil $CO_2$ balance under the other species in drained sites was practically neutral during the study period, pine stands showed relatively high $CO_2$ removals. In the Myrtillus site type nutrient concentrations were, on average, 1.5 to 5.4 times lower compared to Oxalis sites. In addition, Myrtillus sites had significantly lower pH levels and a higher C:N ratio (Figure 2). The distinctive $CO_2$ balance patterns between the site types suggest that, in addition to lower nutrient concentrations, increased soil acidity and differences in soil organic matter and litter input quality (decomposability) had a role in the observed $CO_2$ removals in these sites. Previous studies have also reported a negative correlation between pH and soil C content (Zhou et al., 2019) and have linked soil acidification with increasing soil C stocks (Madsen et al., 2025; Marinos and Bernhardt, 2018). The relevance of soil chemical parameters in determining soil $CO_2$ balance was supported by RDA and PLSR analysis, revealing that soil pH and macronutrient concentrations were key parameters determining soil $CO_2$ balance. Both soil pH and macronutrient concentrations may be influenced by the tree species (Reich et al., 2005; Dawud et al., 2016). Conifers, especially Scots pine, are linked with lowered soil pH (Reich et al., 2005), which may be a way to engineer the ecosystem to its own favour, as Scots pine unlike the other tree species found in our sites can also thrive in nutrient-poor peatland forests (e.g., Ohlson, 1995). The soil acidification in coniferous stands can be attributed to litter quality (Reich et al., 2005; Brock et al., 2019). However, in undrained coniferous stands, soil pH was clearly higher compared to drained sites, and similar to that of undrained broadleaf sites (Figure 2). This is likely because undrained sites may receive more groundwater inputs that neutralize soil acidity than the drained sites. Our drained sites mostly had more acid soils than the undrained ones, a pattern that has previously been recorded for boreal drained peatland forests (Laiho & Laine, 1990). Apart from the reduced groundwater influence, the phenomenon has been explained by a reduction in soil buffering capacity resulting from leaching and tree uptake of base elements such as Ca and Mg, as well as increased oxidation of both organic and inorganic compounds following drainage, contributing to a gradual increase in soil acidity over time (Laine et al., 2006).

Among the drained sites, pine stands had the lowest mean soil $CO_2$ efflux (Figure 6, b). Additionally, nutrient availability correlated negatively with belowground biomass (bGV, FRP) confirming previous observations that greater belowground biomass is associated with reduced nutrient availability (Zhang et al., 2024). Higher $CO_2$ efflux observed in Oxalis sites were consistent with the previous observations of higher organic matter decomposition rates typically observed in sites with high nutrient availability (Hiraishi et al., 2013; Shahbaz et al., 2022). However, an increased total soil $CO_2$ influx from litter was also observed in these sites during the study period, effectively offsetting soil C loss from Rhet. Another potential contributor to the positive $CO_2$ balance in drained soils was soil compaction induced by drainage, as the increased BD of drained soil was found to be associated with lower Rhet emissions. The reason may be reduced soil porosity limiting gas exchange between the soil and the atmosphere (Ball, 2013; Novara et al., 2012). Thus, while drained soil may be prone to a higher decomposition rate, flux-driving processes seem to be countered by increased soil compaction.

In any case, the mean $CO_2$ removals we estimated are uncertain and remain indicative, preventing a definitive conclusion that the soil functions as a $CO_2$ sink. Some uncertainty in the results arises from the inherent variation of study sites categorised into different forest site types and drainage statuses; such variation is natural and cannot be considered erroneous (see, e.g., Westman & Laiho, 2000; Ojanen et al., 2010). However, based on the observed patterns, we consider site stratification by drainage status and site type to be an appropriate approach for interpreting soil $CO_2$ balance. This stratification captures key ecological differences that are relevant to C dynamics and supports meaningful comparisons across site conditions. The results obtained reflect $CO_2$ balances only for the study period, specific to the respective stands in their specific developmental stages and site conditions. They do not represent average $CO_2$ balance over a longer timeframe, such as an entire forest management cycle. In similar conditions, organic soils have been found to be a $CO_2$ source following regenerative felling (Butlers et al.,

2022; Korkiakoski et al., 2023). Therefore, if the impact of management practices were considered, the soil $CO_2$ balance would
likely shift towards reduced $CO_2$ sequestration.
Our $CO_2$ balance estimate does not include the impacts of dissolved organic carbon (DOC) leaching and methane ($CH_4$)
emissions. DOC leaching occurs from both drained and undrained organic soils; however, in drained sites, related C losses
can be increased by 0.43 to 0.78 t C ha$^{-1}$ yr$^{-1}$ (Hiraishi et al., 2013). While $CH_4$ emissions from drained organic soils, including
emissions from drainage ditches in temperate zone, are generally minor, averaging around 5.9 kg $CH_4$-C ha$^{-1}$ year$^{-1}$, C losses
by $CH_4$ emissions from undrained soils are highly variable and uncertain, ranging from 0 to 856 kg $CH_4$-C ha$^{-1}$ year$^{-1}$ (Hiraishi
et al., 2013). Consequently, not accounting for DOC and $CH_4$ when comparing the C balance of drained and undrained organic
soils increases the uncertainty, while these impacts potentially offset each other.
**4.2 Soil $CO_2$ emission factors**
The reported soil $CO_2$ balance values can be directly used as EFs (Table 4). Considering the lack of sufficient evidence for
differences in drained soil $CO_2$ efflux (Table S9) or balance (Figure S4) between countries (see the Supplementary text), the
use of regional soil $CO_2$ EFs is recommended over country-specific values. For example, the mean soil $CO_2$ balance for drained
soils can be treated as EF of $-1.06 \pm 0.45$ t $CO_2$-C ha$^{-1}$ year$^{-1}$, which falls within the range of boreal forest organic soils $CO_2$
EFs reported by previous studies (Table 4, Figure 11). The highest values observed in this study fall within the range reported
in temperate-zone studies. The greater discrepancy with the temperate emission factors may be explained by the limited number
of temperate studies available and their focus on a narrower range of site conditions compared to those represented in the
Baltic countries.
**Table 4: Organic soil $CO_2$ emission factors (t $CO_2$-C ha$^{-1}$ year$^{-1}$) of this study compared to those reported in previous studies.** Values
for the boreal and temperate zones are synthesised from the results of previous studies on nutrient-rich organic soils (Jauhiainen et al., 2023),
while values for the hemiboreal zone reflect the EFs derived in this study. Abbreviations: NR – nutrient rich; NP – nutrient poor; Min and
Max refer to the minimum and maximum values reported; CImin and CImax represent the lower and upper 95% confidence limits for the
mean; N – number of estimates.

| Climate zone | Drainage status | Mean | Min | Max | CImin | CImin | N |
|---|---|---|---|---|---|---|---|
| Boreal | Drained | 0.71 | -3.70 | 7.86 | 0.34 | 1.08 | 103 |
| Hemiboreal | Drained | -1.06 | -5.69 | 2.00 | -1.51 | -0.61 | 19 |
|  | Undrained | -1.27 | -3.80 | 1.35 | -2.00 | -0.54 | 7 |
| Temperate | Drained | 1.61 | -0.08 | 2.93 | 1.15 | 2.07 | 16 |


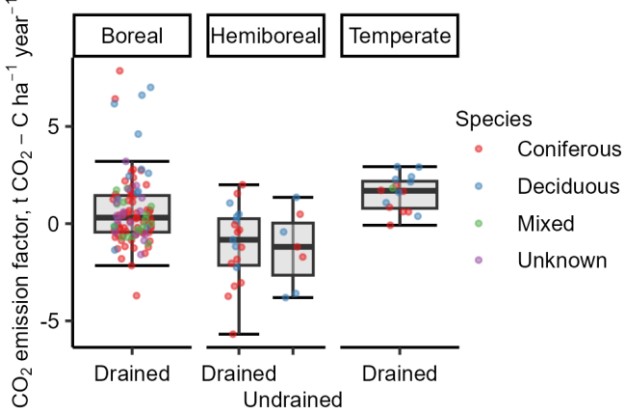


**Figure 11: Organic soil $CO_2$ emission factors of this study compared to those reported in previous studies.** Values for the boreal and
temperate zones are synthesised from the results of previous studies on nutrient-rich organic soils (Jauhiainen et al., 2023), while values
for the hemiboreal zone reflect the EFs derived in this study.

### 4.3 Total respiration

No significant differences in Rtot were observed between the countries, likely because the gradient in the mean air temperature from Estonia to Lithuania, ranging from 6.4 to 8.5 °C, was not substantial enough to introduce distinguishable differences. Nevertheless, while temperature is generally recognised as a strong factor influencing soil respiration and its variation, it should not be regarded as the sole predictor of respiration. Relationships observed in one region may not directly apply to another, as differences in soil moisture (Jovani-Sancho et al., 2018) or nutrient status, as discussed earlier, can significantly alter the respiration dynamics. Similarly, no clear impact of dominant tree species on Rtot was found. This points to a minor role of dominant tree species on emissions. However, there is some evidence that emissions in undrained sites tended to be higher in deciduous stands, particularly alder stands, according to the measured instantaneous emissions. The enhanced soil $CO_2$ efflux observed in the presence of alder can probably be attributed to the symbiotic nitrogen fixation associated with these trees (Warlo et al., 2019), which increases nitrogen availability in the soil. Nitrogen availability, in turn, can stimulate decomposition processes, leading to a higher rate of $CO_2$ release. However, we did not observe increased nitrogen levels in neither the soil nor soil water of alder sites. Although statistically unconfirmed, a tendency can be noticed that in drained sites Rtot emissions tend to be higher in birch stands, but lower in pine forests. Also, previous studies indicated that deciduous stands show higher $CO_2$ emissions (Jauhiainen et al., 2023).

While it was found that both drainage status and WTL threshold above or below 30 cm can be used as a predictor of Rtot, a meaningful correlation between WTL and Rtot was not found. Furthermore, although the absolute variation of the WTL was higher in drained sites, the relative variation in both WTL level and Rtot was indifferent to the drainage status. The observations suggest that higher WTL conditions in undrained sites, while decreasing Rtot emissions, do not guarantee higher resilience to moisture regime disturbances, i.e., more stable emissions. The main reason is that the presence of drainage ditches is not the only factor constraining WTL both spatially and temporally, and in undrained sites too, WTL frequently falls below 30 cm (Butlers et al., 2023) ensuring oxic conditions in soil layers containing labile organic matter. Furthermore, this typically happens in summer (Butlers et al., 2023) when increased temperatures further promote organic matter mineralization. The role of WTL dynamics is reflected also in PCA, showing higher dispersion of drained sites likely due to higher variation in WTL depths. This may be the reason complicating the quantification of relationships between Rtot and the affecting factors, especially in drained sites. The wide range of mean WTL measured at the drained sites also helps explain why Rtot at these sites is not necessarily significantly higher compared to undrained sites.

To achieve accurate Rtot annualization using data from periodic flux measurements, data interpolation through modelling approaches was applied. For Rtot interpolation we compared nonlinear models and linear models after logarithmic or Box-Cox transformation. Both the advantages and shortcomings of these data transformation methods and modelling approaches have been reported in previous studies. (Box and Cox, 1964; Khomik et al., 2009; Liaw et al., 2021; Moulin et al., 2014; Wutzler et al., 2020; Yueqian, 2020). Although the bias in predicted annual Rtot varied among study sites, the overall impact of different flux modelling approaches on estimated mean annual Rtot of drained and undrained sites was minimal. Specifically, the mean bias of results obtained through the implementation of the Box-Cox transformation compared to other approaches was -2±9%. Thus, while such an impact has been observed in previous studies, the skewing of results due to the annualization of respiration was not identified in our study.

### 4.4 Soil heterotrophic respiration

We used a Rhet value derived from Rtot - Rhet' - as the soil $CO_2$ efflux. Rhet` was derived from Rtot empirically using the Rhet/Rs data from a large dataset (Jian et al. 2021), thereby reducing the effect of potential systematic and random errors in individual studies. Such empirical derivation of Rhet has been acknowledged to be an applicable approach and has been utilized in previous studies (Jauhiainen et al., 2019, 2023). To elaborate the recalculation model we used Rhet and Rs values from

the database (Jian, J. et al., 2021) on forest soil flux in the boreal zone, as existing experience suggests that organic soil emissions in hemiboreal forests are more likely to align with boreal rather than temperate conditions (Bārdule et al., 2022; Butlers et al., 2022; Dubra et al., 2023; Heikkinen et al., 2023; Jauhiainen et al., 2023; Krasnova et al., 2019; Lazdiņš et al., 2024). The choice of using only boreal data tends towards estimating higher Rhet`, compared to the use of temperate data, as illustrated in Figure S2. This approach aimed to avoid the underestimation of soil $CO_2$ efflux. The mean share of Rhet acquired using boreal data was 0.65±0.04 while using temperate data - 0,60±0,15, or around 10% difference. Accordingly, the Rhet proportion values we applied were higher than the typically observed range of 0.5 to 0.6 (Bond-Lamberty et al., 2004; Hanson et al., 2000), demonstrating that our approach avoided underestimating Rhet.

The role of ground vegetation autotrophic respiration in Rtot increases with its biomass (Munir et al., 2017). Therefore, the risk of underestimating Rhet by using Rhet' is further reduced because the Rhet/Rs ratio used to recalculate Rtot to Rhet` does not account for the impact of autotrophic respiration from aboveground vegetation, consequently, the approach tends to rather overestimate the Rhet. This aspect should be considered when assessing our results. When estimating the impact of historical drainage on the soil $CO_2$ balance by comparing sites by drainage status, this bias was likely negligible, because the mean ground vegetation biomass did not significantly differ between drained and undrained sites ($\Delta$=0.53 t dm. ha$^{-1}$).

The applicability of the approach is supported by the comparability of the estimated Rhet` values with Rhet reported in previous studies (Bond-Lamberty and Thomson, 2010). We estimated Rhet` of drained soil to be mean 4.30±1.20 t $CO_2$-C ha$^{-1}$ year$^{-1}$, which is slightly higher than both mean Rhet of forest organic soil found in the boreal zone (4.09 t $CO_2$-C ha$^{-1}$ year$^{-1}$) (Ojanen et al., 2010) and in a broader regional scale (3.71±0.53 t $CO_2$-C ha$^{-1}$ year$^{-1}$) (Jian, J. et al., 2021). When attempting to correct for the approximated overestimation of Rhet introduced by trenching (see the Supplementary text), the resulting mean Rhet would be 2.4 t $CO_2$-C ha$^{-1}$ year$^{-1}$, which is considerably lower than the Rhet` values we used in the $CO_2$ balance estimates.

**4.5 Carbon influx by litter inputs**

In the estimation of $CO_2$ influx, we considered data only for fLF, aGV, bGV, and FRP, excluding cLF, MP and dwarf shrubs. This approach was chosen because the $CO_2$ emissions produced in decomposition of these litter types was directly included in the measured Rtot. For instance, cLF due to its dimensions and scarce coverage could not be objectively included in chamber measurements. Furthermore, while fLF is relatively uniform in forest areas, the coverage of mosses and dwarf shrubs is not always so, therefore it is necessary to know their area of projection to be included in the $CO_2$ balance estimation. One solution for incorporating the "missing" litter inputs would be to use modelling approaches (Alm et al., 2023). However, we did not attempt to include those litter sources in the soil $CO_2$ balance estimation, as doing so would have introduced additional uncertainty. We estimated that annual moss production was 22±10% of the average total moss biomass of 0.50±0.09 kg dry matter (dm.) m$^{-2}$ measured for moss patches in our sites. Thus, mosses could potentially provide annual litter inputs reaching up to 0.98±0.25 t dm. ha$^{-1}$ if their cover was 100%. The mean annual cLF was 0.74±0.23 t dm. ha$^{-1}$. Therefore, the litter input estimates (mean 4.70±1.43 t $CO_2$-C ha$^{-1}$ year$^{-1}$) used in the calculation of soil $CO_2$ balance likely led to overestimated soil C loss. This highlights the need to consider all litter input components in further studies, even though that may clearly increase the workload.

**5. Conclusions**

Although all soils in our study sites were classified as nutrient-rich based on forest site type taxonomy, they included a wide variety, ranging from those near the threshold of organic soil definition to soils with deep peat layers. Consequently, the soils exhibited broad variability in pH, macronutrient concentrations, and C:N ratio. That in turn contributed to the observed behaviour of the soils demonstrating both $CO_2$ sink and source dynamics under both drained and undrained conditions. During the study period, drained soils under birch, black alder, and Norway spruce remained $CO_2$ neutral, while in pine stands the

soils were $CO_2$ sinks, presumably due to the significantly lower nutrient availability limiting mineralization of the organic matter. The disparity in soil nutrient conditions also explains why some undrained soils, characterized by relatively high nutrient availability, acted as $CO_2$ sources. These findings highlight the potential to improve predictions of soil $CO_2$ balance by complementing the broad "nutrient-rich" soil classification - typically assigned using site vegetation as a proxy - with quantitative measurements of soil nutrient status.

The study provides a notable contribution through both the plot-level summary and raw data on soil $CO_2$ influx and efflux. The spatial coverage of the study sites, along with the variability in stand characteristics, soil properties, and water-table level dynamics, provides input for synthesising dynamic empirical soil $CO_2$ balance models that depend on drainage status, meteorological conditions, soil chemistry, and stand-related parameters. The reported soil $CO_2$ balance values can be directly used as emission factors. Additional research is still needed to expand the dataset for establishing robust quantitative relationships that can be used to reliably identify and predict whether organic soils function as $CO_2$ sinks, sources, or remain in neutral balance, depending on site-specific conditions and annual weather variations.

**Data availability**

Data used for carbon balance estimations is available at https://doi.org/10.5281/zenodo.14968843

**Author contributions**

KS, JJ, RL, AL and KA developed a harmonised methodology. ABu, DČ, TS and MKS managed and processed the study data. ABu wrote the original manuscript, with significant reviewing contributions from RL. JJ, TS, ABā, IL, VS, HV, IL, AH and AJ provided critical reviews and edits to the manuscript.

**Acknowledgements**

The research is conducted within the framework of the project "Demonstration of climate change mitigation potential of nutrients-rich organic soils in the Baltic States and Finland" (LIFE OrgBalt, LIFE18 CCM/LV/001158). The preparation of the article was supported by Latvia Council of Science national research programme project: "Forest4LV – Innovation in Forest Management and Value Chain for Latvia's Growth: New Forest Services, Products and Technologies" (No.: VPP-ZM-VRIIILA-2024/2-0002).

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
