# Peer review of "1. Soil heterotrophic respiration interpretation issues832 1.1. Soil heterotrophic respiration measurements"

_EGUsphere, 2025_

## Author Response (AR1)

**Edits made in accordance with referee questions/observations and editor review**

When improving the article, we targeted suggestions made by reviewers and the editor. In addition to the requested revisions, we checked the whole text carefully and did some minor linguistic edits and clarifications, as well as checked the consistency in some terms used (e.g., site type short names). We hope that this has further made the manuscript easier to read. No new issues were added in this revision. Please find the edits listed in the tables below.

**Review by editor**

| Suggestion | Edit made |
|---|---|
| Reviewer 1 correctly remarked that the C balance includes also CH4 and DOC. In your response, you indicate that you will mention this in the discussion, but that the study focusses on CO2 fluxes. In addition to mentioning the role of CH4 and DOC, I therefore recommend to also replace C balance/C efflux and alike by CO2 balance/CO2 efflux wherever applicable in the text. | We have added a discussion on DOC and $CH_4$ at the end of the Discussion section 4.1 Soil $CO_2$ balance. We followed the recommendation and replaced "C flux", "C influx", "C efflux" and "C balance" with "$CO_2$ flux", "$CO_2$ influx", "$CO_2$ efflux" and "$CO_2$ balance" throughout the manuscript. |
| Reviewer 2 commented on the arbitrary nature of including country as a factor in your analysis. In your response, you indicate that you are reluctant to follow this suggestion because of the interest in country-based estimates. While I follow this argument, I think this rationale should be better clarified in the manuscript. Perhaps the discussion on the country-based aspects can also be reduced to retain only the most essential parts. | We followed the recommendation and reduced the emphasis on country-based aspects by replacing the stratification by country and drainage status shown in Figures 5 and 6 with stratification by drainage status only. The original figures were moved to the Supplementary Materials, along with corresponding references in the text. As a result, the country-specific focus was reduced, and only the most essential elements were retained to support our assessment that regional emission factors should be used instead of country-specific ones. |

Anonymous Referee #1

| Question/Observation | Edit made |
|---|---|
| The Reference is incomplete on p. 21 | Corrected |
| The distinction of hemi-boreal is a bit confusing with other terms, such as cool temperate, Cool Temperate Mist climate region etc. and could be clarified (says on | To avoid confusion distinguishing hemi-boreal zone and Cool Temperate Moist climate zone we edited sentence in lines 90-93 as follows: |

| | |
|---|---|
| line 50 'between the temperate and boreal zones'). | "This limitation may have hindered the identification of emission-impacting factors and the ability to quantify their relationships, underscoring the need for more localized studies to address these gaps, particularly in the hemiboreal vegetation zone which partly overlaps with the Cool Temperate Moist climate zone (Calvo Buendia et al., 2019) - a subregion of temperate zone as defined by the IPCC." In addition, we standardised the use of terms temperate zone and boreal zone across the manuscript." |
| It appears that there was no attempt (or success) to include a comparison of drained and undrained sites, based on the latitude and longitude data in Table S1, though there appear to be two pairs in Latvia (Fig. 1). Please clarify | We added a sentence to line 132 to explain why we did not aim for or attempt such a pairwise evaluation in this study:

 "All comparisons will be done at group level, since pairwise comparison, even of closely located sites that belong to the same site type, is hampered by the inherent variation in soil characteristics (see Laiho and Pearson 2016)." |
| Limited replication of site types means that categorization of type is unwarranted (lines 536 and following | We disagreed, and revised the paragraph (lines 574-579) as follows:

 "Some uncertainty in the results arises from the inherent variation of study sites categorised into different forest site types and drainage statuses; such variation is natural and cannot be considered erroneous (see, e.g., Westman & Laiho, 2000; Ojanen et al., 2010). However, based on the observed patterns, we consider site stratification by drainage status and site type to be an appropriate approach for interpreting soil $CO_2$ balance. This stratification captures key ecological differences that are relevant to C dynamics and supports meaningful comparisons across site conditions." |
| In the C balance, the expectation is that this measure (C tons balance etc.) converts into $CO_2$. This maybe the case, but what about other C forms in the C cycle? Methane would play an insignificant role in the C balance for most of the sites, given the low water table in most sites, including the undrained ones: probably up to 0.05 t C/ha/yr in the wetter sites and maybe CH4 uptake in the drier sites. Loss of dissolved organic carbon (DOC) would result from leaching of the soil, and may account for up to 0.10 t C/ha/yr additional loss, but also small to most of the soil C balance estimates that have been made. | To address DOC and $CH_4$ we added related discussion at the end of chapter 4.1. Soil carbon balance (last paragraph):

 "Our $CO_2$ balance estimate does not include the impacts of dissolved organic carbon (DOC) leaching and methane ($CH_4$) emissions. DOC leaching occurs from both drained and undrained organic soils; however, in drained sites, related C losses can be increased by 0.43 to 0.78 t C $ha^{-1}$ $yr^{-1}$ (Hiraishi et al., 2013). While $CH_4$ emissions from drained organic soils, including emissions from drainage ditches in temperate zone, are generally minor, averaging around 5.9 kg $CH_4$-C $ha^{-1}$ $year^{-1}$, C losses by $CH_4$ emissions from undrained soils are highly variable and uncertain, ranging from 0 to 856 kg $CH_4$-C $ha^{-1}$ $year^{-1}$ (Hiraishi et al., 2013). Consequently, not accounting for DOC and $CH_4$ when comparing the C balance of drained and undrained organic soils increases the uncertainty, while these impacts potentially offset each other." |
| The manuscript started with a comment on the use of Emission Factors by the IPCC and states, though no EF values were given. | We noted that our C balance results can be used as EFs and illustrated comparability with EFs reported by previous studies by |

| | |
|---|---|
| If the objective of the study, beyond the science of the forested systems, was to contribute to a better estimate of the variability and magnitude of EF, it would be useful to see how the authors think these study would contribute to that objective. What 'better' estimate of EF could have been made using the results assembled in the manuscript, with a lot of good, hard work over two years and standardized methods, compared to the 'guesswork' of the past? | supplementing discussion by introducing short, dedicated section 4.2 Soil $CO_2$ emissions factors:

"The reported soil $CO_2$ balance values can be directly used as EFs (Table 4). Considering the lack of sufficient evidence for differences in drained soil $CO_2$ efflux (Table S9) or balance (Figure S4) between countries (see the Supplementary text), the use of regional soil $CO_2$ EFs is recommended over country-specific values. For example, the mean soil $CO_2$ balance for drained soils can be treated as EF of $-1.06 \pm 0.45$ t $CO_2$-C ha$^{-1}$ year$^{-1}$, which falls within the range of boreal forest organic soils $CO_2$ EFs reported by previous studies (Table 4, Figure 11). The highest values observed in this study fall within the range reported in temperate-zone studies. The greater discrepancy with the temperate emission factors may be explained by the limited number of temperate studies available and their focus on a narrower range of site conditions compared to those represented in the Baltic countries." |
| 24 It seems that the estimated changes in C do not involve + and – signs. Such as soil C removal from drained Scots pine sites was 2.77 units while C sink occurred in undrained black alder sites there was an average sink of 1.33 units. Throughout the manuscript could 'loss' estimates be given a negative sign (e.g. -2.77 +/- 0.36 units) and 'gain' estimates be given a positive sign (1.33 +/- 0.72 units. The graphs showing the 'C balance' (Fig 9 and 10) include negative values, please be consistent. The notation used in Figures also varies: for example Fig 8 has 'Carbon flux' and 9 and 10 have 'C balance' with the same units and meaning. Please standardize. | We standardized the signs across the manuscript as suggested, adding – for values indicating soil C loss and + for values indicating soil C increase.

For Figures 9 and 10 we noted in caption that positive values indicate soil C stock increase. For figure 8 we further noted that efflux indicates soil C losses and influx soil C gains. |
| 15 the boreal region | Added "the" |
| 35 why not use 'faster' rather than 'higher' to describe a rate? | Revised to faster. |
| 46 One of the studies was on a drained peatland used for horticultural crop production, so is not representative of the types used in the EF estimates. | We retained these references, as they are cited as sources used in the development of the IPCC default Tier 1 emission factors. We double checked https://doi.org/10.1029/93GB00469, both crop and forest covers are scope of the article. We made no edits here. |

| | |
|---|---|
| 83 Jauhianen et al. was incompletely cited in the References. | Revised |
| 164 how 'small' was insignificant? | We replaced "insignificant" with "small (<5%)". |
| 460 Basal area had the strongest correlation with C balance, yet in Fig. 10, the R2 of 0.14 was the smallest in the 6 graphs, several with p values < 0.01. Please check. It would be good to include the slope of the regression to indicate how much change in C balance was created by a change in the independent variable. For example, a reduction in pH from 6 to 2.5 (!) would result in a C balance (gain) of about 3 units. An increase in bGV of 0.1 to 3.1 units would result in a C balance of -1 to 4 units. | Since basal area showed the strongest correlation among the tree stand variables, it is the only stand variable presented in Figure 10. Therefore, the corresponding R² value does not have to be the highest, compared to the other variables presented in the figure. We revised "stand" to "tree stand" to make this clearer. We added the slope values. |

**#2 Jens-Arne Subke**

| Question/Observation | Edit made |
|---|---|
| Emission Factors form a significant part of the rationale of the study, taking up several paragraphs in the introduction. This is not reflected in the discussion which focusses much more on fundamental understanding of C balances, not emissions reporting. It would be better to address this by setting the scope of the study and motivation for study differently in the introduction. | Addressed by introducing a dedicated short section 4.2 Soil $CO_2$ emission factors in the Discussion, and editing Conlusions to highlight the article's contribution to the development of emission factor. |
| Soil pH is cited throughout the manuscript as an important correlator of C balance. It is generally presented as a causal link of lower pH and C stocks. However, the cause of pH differences are not considered meaningfully, where conifer plantations are likely to have reduced pH due to acidic litterfall. The correlation between C stocks and pH are hence linked to vegetation more than pH being an independent driver of C stocks. This should be much clearer in the discussion (e.g. 520-525). | We complemented discussion in lines 551-562: "Both soil pH and macronutrient concentrations may be influenced by the tree species (Reich et al., 2005; Dawud et al., 2016). Conifers, especially Scots pine, are linked with lowered soil pH (Reich et al., 2005), which may be a way to engineer the ecosystem to its own favour, as Scots pine unlike the other tree species found in our sites can also thrive in nutrient-poor peatland forests (e.g., Ohlson, 1995). The soil acidification in coniferous stands can be attributed to litter quality (Reich et al., 2005; Brock et al., 2019). However, in undrained coniferous stands, soil pH was clearly higher compared to drained sites, and similar to that of undrained broadleaf sites (Figure 2). This is likely because undrained sites may receive more groundwater inputs that neutralize soil acidity than the drained sites. Our drained sites mostly had more acid soils than the undrained ones, a pattern that has previously been recorded for boreal drained peatland forests (Laiho & Laine, 1990). Apart from the reduced |

| | groundwater influence, the phenomenon has been explained by a reduction in soil buffering capacity resulting from leaching and tree uptake of base elements such as Ca and Mg, as well as increased oxidation of both organic and inorganic compounds following drainage, contributing to a gradual increase in soil acidity over time (Laine et al., 2006)." |
|---|---|
| There is also an apparent mismatch between opening arguments of conducting this study across the three Baltic states, as they share an ecoclimatic region. I agree, but found the partial focus in the analysis to separate results by country unhelpful. This is strongly biased by the distribution of vegetation and drainage across the study, resulting in limited insights, The presentation of data can be streamlined significantly by removing the "country" aspect throughout. | We reduced country aspect by moving 2 figures (panels a in Figures 5 and 6) and related text to the Supplement, while retaining most crucial demonstration that emissions from comparable sites do not significantly differ between countries to provides scientific substantiation that for comparability of GHG inventories in Baltic states, use of different C balance estimation methods/emissions factors are likely inappropriate. |
| 26-28: The past two sentence should be merged. What you say seems to contradict the previous statements where source and sink behaviours are presented as functions of drainage and tree species. Make it clear how different parameters influence carbon balances without causing contradictions. | We merged 2 last sentences as follows: "Variation in the soil $CO_2$ balance was related to soil macronutrient concentrations and pH: forest types characterized by lower nutrient availability showed greater soil $CO_2$ sink." |
| to emphasize 105: Why does the analysis distinguish sites by country? The argument presented is that this is one ecoclimatic zone with site replication across the three Baltic states. From that rationale, national boundaries are arbitrary and the analysis should focus on environmental drivers of biogeochemical patterns. This focusses analysis and removes part of detailed results/discussion. | We reduced the country aspect as explained above. |
| 300 (Fig. 2): I am worried by the confounding effect of drainage and tree species. Table 1 indicates that undrained spruce forests were sampled, but this figure shows only deciduous forest on undrained sites. Looking at pH in particular, the observed difference ascribed to drainage status is caused by undrained sites not including coniferous forest with more acidic litter input. Comparing birch and alder forests only, there is no evident difference by drainage. Looking at Fig. 10, the pH distribution by species and drainage seem to be different to what is shown in Fig. 2 (e.g., drained/Birch has values below 4 in Fig 10, not in Fig. 10; | Figure 2 was updated to correct error. To clarify why Figures 2 and 10 do not have to match, we clarified in the captions that Figure 2 presents results at the subplot level to fully reflect the observed variation, while Figure 10 shows site-level mean values. |

| | |
|---|---|
| undrained Spruce pH values shown in Fig. 10, not Fig. 2). This has to be clarified, as the discussion has to take account of these patterns and potential confounding influences. | |
| 398: This is not evident from Table 3. aGV is c. 39% of total GV (sum of aGV and bGV) in drained, and c. 49% of total in undrained. | We clarified (line 417) that the results are derived from biomass comparisons at the subplot level. |

**Answer to Anonymous Referee #1**

Thank you very much for the thoughtful and thorough review. Your detailed feedback is greatly appreciated and will help us improve the manuscript. Please find responses to questions or observations made below.

| Question/Observation | Response |
|---|---|
| The Reference is incomplete on p. 21 | Will be revised. |
| The distinction of hemi-boreal is a bit confusing with other terms, such as cool temperate, Cool Temperate Mist climate region etc. and could be clarified (says on line 50 'between the temperate and boreal zones'). | To avoid confusion distinguishing hemi-boreal zone and Cool Temperate Moist climate zone we will edit sentence in L86-89 as follows: "This limitation may have hindered the identification of emission-impacting factors and the ability to quantify their relationships, underscoring the need for more localized studies to address these gaps, particularly in the hemiboreal vegetation zone which partly overlaps with the Cool Temperate Moist climate region (Calvo Buendia et al., 2019) - a subregion of temperate zone as defined by the IPCC." We mention both the Cool Temperate Moist climate region and the hemiboreal vegetation zone because the former, which refers to a broader geographical area, is used in national GHG inventories to classify the Baltic States per the IPCC climate zone definitions. In contrast, the hemiboreal zone more accurately reflects the local conditions within the Baltic region. We consider it important to mention both, as the IPCC climate classification is particularly relevant for readers familiar with national GHG reporting, while the term hemiboreal vegetation zone is more commonly used within the scientific community. Highlighting this overlap helps to describe the conditions in which the Baltic States are situated for a broader audience. |
| It appears that there was no attempt (or success) to include a comparison of drained and undrained sites, based on the latitude and longitude data in Table S1, though there appear to be two pairs in Latvia (Fig. 1). Please clarify | We did not aim for or attempt such a pairwise evaluation in this study. The mere closeness of undrained and drained sites does not imply that they could be compared as a pair, even if they represented the same site type when the drained site was still undrained. As shown by our results, and some previous studies, there is |

| | variation in both soil conditions and greenhouse gas emissions among individual sites belonging to the same sites types, making pairwise comparisons questionable. Site type level comparisons are more justified. We will clarify this in the Material and Methods section. |
|---|---|
| Limited replication of site types means that categorization of type is unwarranted (lines 536 and following | We would like to clarify the interpretation of lines 536 and following. In the corresponding paragraph, we acknowledge the observed variability in soil carbon stock balances; however, based on the evidence, we consider categorization by site type to be an appropriate and ecologically meaningful approach. This assessment is supported by two key observations: first, the site types exhibit distinct and consistent differences in soil properties, as shown in Figure 2, which justifies stratification from an ecological perspective; second, this stratification is further supported by the results—on average, nutrient-richer sites acted as carbon sources, while nutrient-poorer sites showed carbon balanced around equilibrium on average, as illustrated in Figure 9c. To include the clarification in the article, we will revise the sentence on line 537 as follows: "Some uncertainty in the results arises from the inherent variation of study sites categorized into different forest site types and drainage statuses; such variation is natural and cannot be considered erroneous (see, e.g., Westman & Laiho 2000, https://doi.org/10.1023/A:1023348806857, and Ojanen et al. 2010, https://doi.org/10.1016/j.foreco.2010.04.036). However, based on the observed patterns, we consider site stratification by drainage status and site type to be an appropriate approach for interpreting soil C balance. This stratification captures key ecological differences that are relevant to C dynamics and supports meaningful comparisons across site conditions." |
| In the C balance, the expectation is that this measure (C tons balance etc.) converts into CO2. This maybe the case, but what about other C forms in the C cycle? Methane | We agree that methane emissions and carbon leaching, while scientifically relevant, contribute marginally to the overall soil carbon balance. However, the scope of this |

| | |
|---|---|
| would play an insignificant role in the C balance for most of the sites, given the low water table in most sites, including the undrained ones: probably up to 0.05 t C/ha/yr in the wetter sites and maybe CH4 uptake in the drier sites. Loss of dissolved organic carbon (DOC) would result from leaching of the soil, and may account for up to 0.10 t C/ha/yr additional loss, but also small to most of the soil C balance estimates that have been made. | article is intentionally limited to the soil carbon balance estimation evaluating direct $CO_2$ emissions as an efflux, as described in the methods section. We will add a short mention on the roles of CH4 and DOC in the Discussion. |
| The manuscript started with a comment on the use of Emission Factors by the IPCC and states, though no EF values were given. If the objective of the study, beyond the science of the forested systems, was to contribute to a better estimate of the variability and magnitude of EF, it would be useful to see how the authors think these study would contribute to that objective. What 'better' estimate of EF could have been made using the results assembled in the manuscript, with a lot of good, hard work over two years and standardized methods, compared to the 'guesswork' of the past? | Thank you for emphasizing the potential to elaborate further on the article's contribution to greenhouse gas emission estimates. To address this, we will expand the final paragraph of Section 3.1 Soil Carbon Balance by noting that the reported soil carbon balance values can be used directly as EFs, and compare them to the IPCC default EFs. We were a bit shy initially because these results reflect carbon balances only for the study period and are specific to the respective stands in their particular developmental stages and site conditions. They do not represent average changes in soil carbon stocks over longer timeframes, such as an entire forest management cycle. However, that is admittedly the case with all static EFs currently. We will clarify that, emphasizing that the study provides a notable contribution through both the plot-level summary and raw data on soil carbon influx and efflux. The spatial coverage of the study sites, along with the variability in stand characteristics, soil properties, and water table level dynamics, provides input for synthesising dynamic empirical soil carbon balance models that depend on drainage status, meteorological conditions, soil chemistry, and stand-related parameters. |
| 24 It seems that the estimated changes in C do not involve + and – signs. Such as soil C removal from drained Scots pine sites was 2.77 units while C sink occurred in undrained black alder sites there was an average sink of 1.33 units. Throughout the manuscript could 'loss' estimates be given | We apologize for the inconsistency and confusion and will standardize these. |

| | |
|---|---|
| a negative sign (e.g. -2.77 +/- 0.36 units) and 'gain' estimates be given a positive sign (1.33 +/- 0.72 units. The graphs showing the 'C balance' (Fig 9 and 10) include negative values, please be consistent. The notation used in Figures also varies: for example Fig 8 has 'Carbon flux' and 9 and 10 have 'C balance' with the same units and meaning. Please standardize. | |
| 15 the boreal region | Will be revised. |
| 35 why not use 'faster' rather than 'higher' to describe a rate? | Will be revised to faster. |
| 46 One of the studies was on a drained peatland used for horticultural crop production, so is not representative of the types used in the EF estimates. | We have to retain these references, as they are cited as sources used in the development of the IPCC default Tier 1 emission factors. We double checked https://doi.org/10.1029/93GB00469, bot crop and forest covers are scope of the article. |
| 83 Jauhianen et al. was incompletely cited in the References. | Will be revised. |
| 164 how 'small' was insignificant? | We will replace "insignificant" with "small (<5%)". |
| 460 Basal area had the strongest correlation with C balance, yet in Fig. 10, the R2 of 0.14 was the smallest in the 6 graphs, several with p values < 0.01. Please check. It would be good to include the slope of the regression to indicate how much change in C balance was created by a change in the independent variable. For example, a reduction in pH from 6 to 2.5 (!) would result in a C balance (gain) of about 3 units. An increase in bGV of 0.1 to 3.1 units would result in a C balance of -1 to 4 units. | Since basal area showed the strongest correlation among the stand variables, it is the only stand variable presented in Figure 10. Therefore, the corresponding $R^2$ value does not have to be the highest. We will add slope values. |

**Answer to Jens-Arne Subke**

Thank you for your comments and the generally positive assessment of the revised manuscript. We appreciate your thorough cross-checking and the suggested refinements, which we will implement. Please find our responses to the questions and observations in the table below.

| Question/Observation | Response |
|---|---|
| The presented study has a somewhat imbalanced representation of tree species, drainage and soil types so that results have to be considered with some caution. | We agree that the distribution of dominant tree species, drainage status, and soil types is important. Accordingly, we present the results stratified by drainage status, tree species, and nutrient status (e.g. Fig 9). |
| Emission Factors form a significant part of the rationale of the study, taking up several paragraphs in the introduction. This is not reflected in the discussion which focusses much more on fundamental understanding of C balances, not emissions reporting. It would be better to address this by setting the scope of the study and motivation for study differently in the introduction. | The motivation of the study is indeed rooted in the necessity to improve the accuracy of organic soil GHG gas inventories in the Baltic states. We were shy of using the term EF in regard to our soil balance results; however, as also Referee #1 called attention to this, we will revise the Discussion with a note that the results can be used as EFs, and compare them to the IPCC default EFs. However, we still consider that the discussion focusing on the understanding of C balances is also warranted because EFs are not separate from these fundamentals; rather, they are derived from and guided by them. |
| Soil pH is cited throughout the manuscript as an important correlator of C balance. It is generally presented as a causal link of lower pH and C stocks. However, the cause of pH differences are not considered meaningfully, where conifer plantations are likely to have reduced pH due to acidic litterfall. The correlation between C stocks and pH are hence linked to vegetation more than pH being an independent driver of C stocks. This should be much clearer in the discussion (e.g. 520-525). | We agree that soil pH can be influenced by the dominant tree species, as well as the drainage status as such. We will address this in the discussion (after line 525) We would, however, like to remain cautious here as in undrained soils pH was not lowered by conifers (please see specific comment on 300). |
| There is also an apparent mismatch between opening arguments of conducting this study across the three Baltic states, as they share an ecoclimatic region. I agree, but found the | We agree that the study material is strongly imbalanced among the countries and thus, the insights concerning the country aspect remain limited. However, we would like to |

| partial focus in the analysis to separate results by country unhelpful.

This is strongly biased by the distribution of vegetation and drainage across the study, resulting in limited insights, The presentation of data can be streamlined significantly by removing the "country" aspect throughout. | retain the country aspect, as demonstrating that emissions from comparable sites do not significantly differ between countries provides scientific substantiation that for comparability of GHG inventories in Baltic states, use of different C balance estimation methods are likely inappropriate. For this reason, we believe that retaining the country-level comparison adds value, even though it is not the most interesting or relevant part of the paper. |
|---|---|
| 26-28: The past two sentence should be merged. What you say seems to contradict the previous statements where source and sink behaviours are presented as functions of drainage and tree species. Make it clear how different parameters influence carbon balances without causing contradictions. | We will thoroughly edit the last two sentences and will carefully check the potential contradictions while doing so. |
| to emphasize 105: Why does the analysis distinguish sites by country? The argument presented is that this is one ecoclimatic zone with site replication across the three Baltic states. From that rationale, national boundaries are arbitrary and the analysis should focus on environmental drivers of biogeochemical patterns. This focusses analysis and removes part of detailed results/discussion. | We fully agree that the national boundaries are arbitrary in this respect, but we need to deal with the country aspect to support the use of the same EFs across the region. The analysis distinguishes sites by country to support the argument presented. Different readers may have different expectations for the paper. We are sorry that the hard work of the referees is increased by the addition of the country aspect, but the need to deal with it comes from the regulations of the GHG inventories. |
| 300 (Fig. 2): I am worried by the confounding effect of drainage and tree species. Table 1 indicates that undrained spruce forests were sampled, but this figure shows only deciduous forest on undrained sites. Looking at pH in particular, the observed difference ascribed to drainage status is caused by undrained sites not including coniferous forest with more acidic litter input. Comparing birch and alder forests only, there is no evident difference by drainage.

Looking at Fig. 10, the pH distribution by species and drainage seem to be different to | Thank you for noticing this issue. It concerns Figure 2, where data from multiple sites - including spruce in undrained conditions - were missing. This omission does not affect any other results or figures in the article. We will update the figure. The inclusion of undrained spruce sites do not lower the mean pH in undrained conditions, as the mean pH in undrained spruce sites was 5.14. This may provide additional explanation why mean soil emissions in undrained sites were not significantly lower than in drained ones - we will add this to the discussion. In the undrained sites, |

| | |
|---|---|
| what is shown in Fig. 2 (e.g., drained/Birch has values below 4 in Fig 10, not in Fig. 10; undrained Spruce pH values shown in Fig. 10, not Fig. 2). This has to be clarified, as the discussion has to take account of these patterns and potential confounding influences. | especially, soil pH may be regulated by the inflowing water.

Please note that the data points in Figures 2 and 10 do not have to match. Figure 2 presents results at the subplot level to fully reflect the observed variation, while Figure 10 shows site-level mean values. We will further clarify this in the captions. |
| 398: This is not evident from Table 3. aGV is c. 39% of total GV (sum of aGV and bGV) in drained, and c. 49% of total in undrained. | The results in line 398 and Table 3 do not have to match. The values reported in line 398 are based on comparisons using higher-granularity data at the subplot level, whereas Table 3 presents average values aggregated by drainage status. To avoid confusion, we will further clarify in line 398 that the results are derived from biomass comparisons at the subplot level. |

---

## Author Response (AR2)

**Author's Response**

Thank you for the suggestions. We made final corrections and edits accordingly:

- The title of the paper is renamed.
- 2 sentences are added to the abstract:
  - Irrespective of drainage status, the soils functioned as both $CO_2$ sinks and sources (L23).
  - The reported soil $CO_2$ balance values may be used as regional emission factors (L29).
- "S" added to the numbering of sections of supplementary text.